# An ambruticin-sensing complex modulates *Myxococcus xanthus* development and mediates myxobacterial interspecies communication

Francisco Javier Marcos-Torres[1,3], Carsten Volz [1,3] & Rolf Müller [1,2✉]

Starvation induces cell aggregation in the soil bacterium *Myxococcus xanthus*, followed by formation of fruiting bodies packed with myxospores. Sporulation in the absence of fruiting bodies can be artificially induced by high concentrations of glycerol through unclear mechanisms. Here, we show that a compound (ambruticin VS-3) produced by a different myxobacterium, *Sorangium cellulosum*, affects the development of *M. xanthus* in a similar manner. Both glycerol (at millimolar levels) and ambruticin VS-3 (at nanomolar concentrations) inhibit *M. xanthus* fruiting body formation under starvation, and induce sporulation in the presence of nutrients. The response is mediated in *M. xanthus* by three hybrid histidine kinases (AskA, AskB, AskC) that form complexes interacting with two major developmental regulators (MrpC, FruA). In addition, AskB binds directly to the *mrpC* promoter in vitro. Thus, our work indicates that the AskABC-dependent regulatory pathway mediates the responses to ambruticin VS-3 and glycerol. We hypothesize that production of ambruticin VS-3 may allow *S. sorangium* to outcompete *M. xanthus* under both starvation and growth conditions in soil.

[1] Department of Microbial Natural Products (MINS), Helmholtz Institute for Pharmaceutical Research Saarland (HIPS), Helmholtz Centre for Infection Research (HZI), Campus E8 1, 66123 Saarbrücken, Germany. [2] German Center for Infection Research (DZIF), Braunschweig 38124, Germany. [3] These authors contributed equally: Francisco Javier Marcos-Torres, Carsten Volz. ✉email: rolf.mueller@helmholtz-hips.de

Besides their large potential to produce bioactive compounds, myxobacteria are excellent model organisms to study cell-to-cell signaling and gene regulation during a complex life cycle involving multicellular development. These features have been thoroughly studied using *Myxococcus xanthus* DK1622 but are still far from being comprehensively understood. Under starvation, cells form multicellular fruiting bodies (FB) in a synchronized and tightly regulated process. Inside FB, a portion of vegetative cells differentiates into spherical myxospores resistant to various adverse conditions including sonication[1,2].

Development is coordinated by tightly regulated signaling cascades controlling the sequential expression of more than 2000 genes during the formation of FB and sporulation[3]. The known gene regulatory network (GRN) initially reacts to increasing intracellular levels of (p)ppGpp[4–7]. Subsequent signals known as the A, E, and C-signals coordinate the developmental program required for aggregation and sporulation[1,8]. During development, the transcriptional regulators FruA and MrpC play a key role in controlling the transcription of hundreds of developmental genes[8–10].

Nevertheless, *M. xanthus* is also able to sporulate even under nutrient-rich conditions. This starvation-independent sporulation is achieved in the presence of 0.5 M glycerol in liquid cultures[11]. Other chemicals are also able to trigger this effect. However, all chemical inducers require concentrations unlikely to occur in nature[12,13]. The reason for *M. xanthus* to exhibit a mechanism responding to likely unphysiological amounts of such effectors remains elusive. Even though the effects of the chemical inducers have been studied to some extent, it is unknown whether these effectors influence development under starvation conditions.

Motivated by observing a striking inability of *M. xanthus* to form FB when co-cultivated with *Sorangium cellulosum*, we initiated research to identify the factor causing this interspecies interaction and here provide evidence that the antifungal ambruticin VS-3 is a "natural" chemical inducer of sporulation. We show that ambruticin VS-3 also acts as inhibitor of FB formation and starvation-induced sporulation at low nanomolar concentrations. We demonstrate that these effects can also be induced by glycerol, the well-known, but seven orders of magnitude less potent inducer. Furthermore, we identify three interacting kinases, AskA, AskB, and AskC, as one direct entry point of the ambruticin VS-3/glycerol cascade, which are also shown to interact with MrpC and FruA in vitro. Finally, we demonstrate that those interacting proteins influence the established regulatory cascades by modulating the activity of the promoter $P_{mrpC}$. This could be shown by both, the altered binding of MrpC in the presence of ambruticin VS-3 and kinase AskB that binds preferentially to DNA comprising promoter $P_{mrpC}$ in vitro.

## Results

**Ambruticin VS-3 acts as both, potent inhibitor of development and chemical inducer of sporulation.** *M. xanthus* is exposed to a highly variable habitat where the availability of nutrients and an effective response to environmental substances are crucial for survival. The ecological niche certainly involves interaction with competing microorganisms like other myxobacteria. To date, such interactions have not been investigated in detail. To obtain insights into such interspecies interactions, we performed co-cultivations using various myxobacteria. One co-cultivation using *Sorangium cellulosum* So ce10[14] and *M. xanthus* DK1622 attracted our attention: In contrast to regular FB development, cells of *M. xanthus* spotted close to a colony of So ce10 did not form FB (Fig. 1a). No contact between both colonies was required for the inhibition. Upon co-cultivation of *M. xanthus* together with other myxobacteria no such effect could be observed.

*S. cellulosum* is known to be a rich source of secondary metabolites the natural function of which are largely unknown. Thus, they are attractive candidates for interspecies interaction. To identify the effector of the inhibition, known and purified secondary metabolites produced by So ce10 were tested against *M. xanthus* under developmental conditions. Strikingly, the development of *M. xanthus* was inhibited in the presence of as low as 4 nM ambruticin VS-3 (Fig. 1c), a polyketide that is known for its antifungal activity (Fig. 1b)[15]. This observation, together with the fact that none of the additional myxobacteria used in the co-cultivations are able to produce ambruticin VS-3 strongly suggested that ambruticin VS-3 is the causative agent of this effect on FB formation in *M. xanthus*. The extracellular concentration of ambruticin VS-3 during cultivation of So ce10 in the experiments was determined to range from 0.4 μM ± 0.1 μM to 4 μM ± 1 μM (24 h to 120 h of incubation, respectively; Fig. 1b). Further phenotypical analysis of the colonies revealed that ambruticin VS-3 also inhibits completion of sporulation rendering shortened vegetative cells, which present a thickened cell wall similar to that of the mature myxospores (Fig. 1e). Upon sonication, those "pseudospores" were resistant to disruption to a similar extent as the round-shaped myxospores produced under starvation conditions. Nevertheless, the pseudospores were severely impaired in germination (Fig. 1f).

Having observed this remarkable effect of ambruticin VS-3 on FB formation we wondered if ambruticin VS-3 might also affect growth in nutrient-rich CTT medium. Indeed, when ambruticin VS-3 was added, spore-like cells could be observed (Fig. 1g). This starvation-independent sporulation is known to occur in the presence of glycerol and has been thoroughly studied[11–13]. However, glycerol-induced sporulation is observed at millimolar concentrations, which are unlikely to occur under environmental conditions (Fig. 1g). In contrast, the ambruticin-induced sporulation only requires nanomolar concentrations. The resulting spores were found to germinate with similar efficiency as glycerol-induced spores (Fig. 1h).

As we could observe similar effects in presence of ambruticin VS-3 and glycerol under nutrient-rich conditions, we aimed to elucidate if the addition of glycerol to starvation medium would also result in an inhibitory effect on *M. xanthus* development like observed for ambruticin VS-3. Indeed, glycerol is able to inhibit FB formation and sporulation in *M. xanthus* (Fig. 1d, f) suggesting a similar mechanism of action of both chemical inducers. Again, glycerol acts at millimolar concentration in contrast to nanomolar concentrations of ambruticin VS-3.

The difference in the effective concentration of both compounds, together with the likelihood of natural co-occurrence of *M. xanthus* and *S. cellulosum* might indicate ambruticin VS-3 is a natural trigger of chemically induced sporulation and an inhibitor of starvation-induced development. Strain So ce10 produces sufficient amounts of ambruticin VS-3. The production titer of ambruticin VS-3 significantly exceeds the minimal concentration needed for inhibition of FB formation when using purified ambruticin VS-3 (see above). It is tempting to speculate that by the action of ambruticin VS-3, strains producing ambruticin VS-3 like So ce10 are able to outcompete "sensitive" strains like *M. xanthus* in the soil environment. Moreover, this would be true for both starvation and during growth in the presence of nutrients. Under starvation *M. xanthus* would be unable to sporulate efficiently due to the abolished FB formation. Under growth conditions, *M. xanthus* cells would be forced to differentiate into pseudospores what would prevent *M. xanthus* from propagating.

**Three interacting hybrid histidine kinases are involved in sensing ambruticin VS-3 and glycerol.** Good candidate sensors

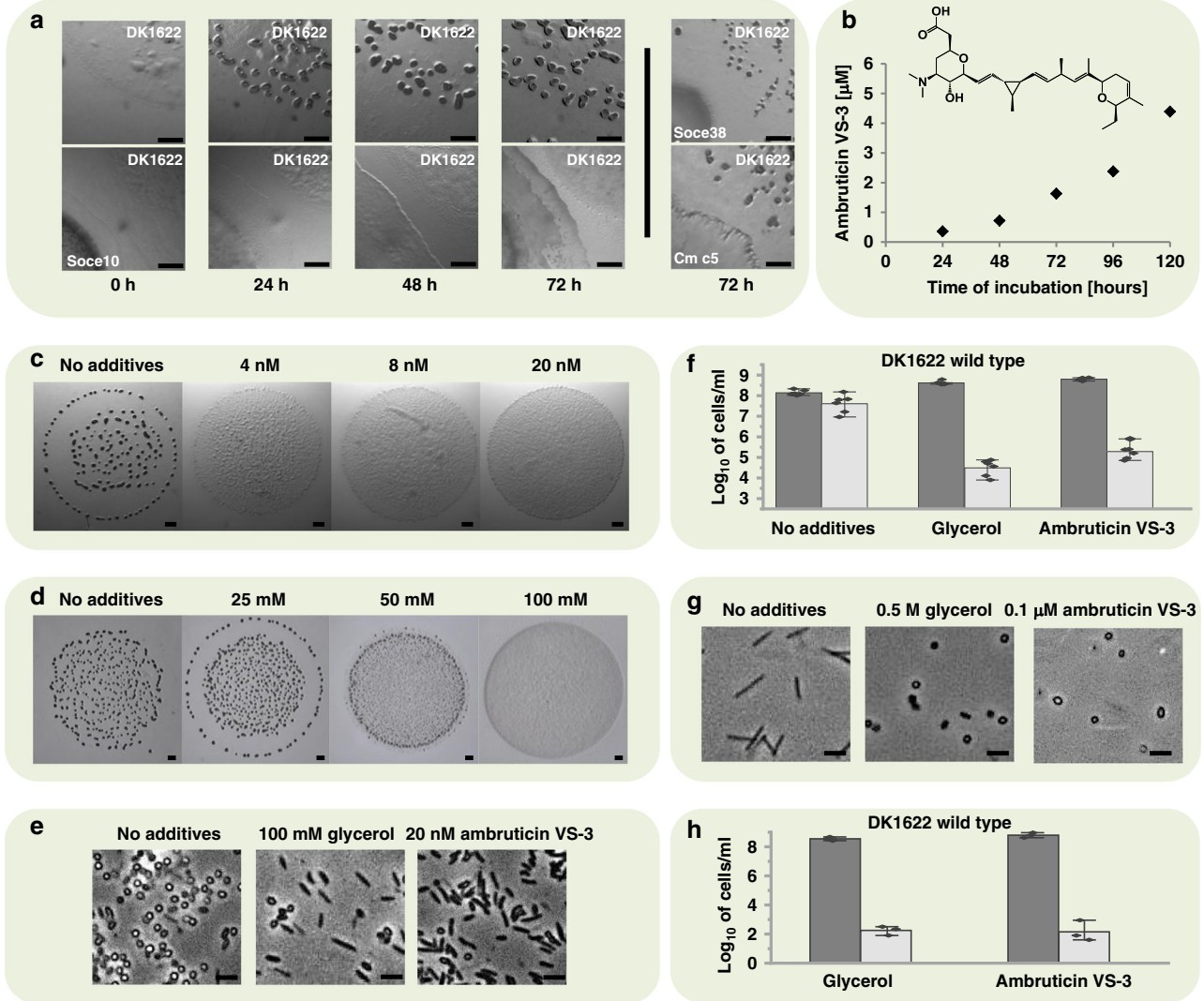

**Fig. 1 Phenotypical effects on *M. xanthus* upon co-cultivation with *Sorangium cellulosum* or addition of ambruticin VS-3 and glycerol. a** Developmental monitoring of *M. xanthus* DK1622 on starvation agar in single cultivation (upper panel) or in co-cultivation with the ambruticin producer *S. cellulosum* Soce10 (lower panel). On the right side, two representative co-cultivation attempts of *M. xanthus* DK1622 and two strains not producing ambruticin VS-3 are shown after 72 h (*Sorangium cellulosum* Soce38 and *Chondromyces crocatus* Cm c5). Scale bars represent 0.5 mm. Pictures were taken at the time points indicated below each photograph. **b** Extracellular concentration of ambruticin VS-3 during the growth of *Sorangium cellulosum* So ce10. The time-dependent concentrations were determined based on two independent growth experiments. The 2D structure of ambruticin VS-3 with the molecular formula $C_{30}H_{47}NO_5$ is shown on the top. FB formation of *M. xanthus* DK1622 on CF starvation medium with different concentrations of ambruticin VS-3 **c** or glycerol **d**. Pictures were taken after 48 h of incubation. Scale bars represent 0.5 mm. **e** Spores and pseudospores produced during FB formation in presence of glycerol or ambruticin VS-3 during development. Pictures were taken after 48 h. Scale bars represent 1 μm. 20 nM ambruticin VS-3 or 100 mM glycerol were used in this experiment as fruiting body formation was completely abolished at these concentrations. The experiments were performed in triplicates. **f** Number of sonication resistant pseudospores after addition of 60 nM ambruticin VS-3 or 300 mM glycerol to developing cells (dark gray bars) and germination efficiency (light gray bars) after 96 h of development. $N = 3$ biologically independent samples. The data are presented as mean values ± SEM. **g** Chemically induced spores under nutrient-rich conditions by addition of 0.5 mM glycerol and 0.1 μM ambruticin VS-3. Pictures were taken 6 h after induction of sporulation. Scale bars represent 1 μm. As on nutrient-rich medium, a five times higher concentration of glycerol is needed to induce sporulation, the concentration of ambruticin VS-3 has been adjusted accordingly. The experiments were performed in triplicates. **h** Number of sonication resistant cells 6 h after chemical induction by the use of 100 nM ambruticin VS-3 or 500 mM glycerol under growth conditions (dark gray bars) and germination efficiency (light gray bars) after 96 h of development. $N = 3$ biologically independent samples. The data are presented as mean values ± SEM. Source data are provided as a Source Data file.

for compounds like ambruticin VS-3 are classical two component systems (TCS), which are the most abundant signal transduction mechanisms encoded in *M. xanthus*[16]. In a simple TCS, a histidine kinase senses a signal followed by auto-phosphorylation and transfer of the phosphate to the response regulator. Finally, transcription of target genes is regulated as a response[17]. To connect ambruticin VS-3 to some potential regulatory and

sensing proteins, we turned our attention to the producer of ambruticin VS-3 and analyzed the biosynthetic gene cluster of *S. cellulosum* So ce10. Indeed, it encodes a hybrid histidine kinase (HHK; Amb07) resembling members of the HHK group III[15]. Such group III HHK are signaling proteins containing multiple HAMP domains (poly-HAMP), a histidine kinase domain and several receiver domains[18,19]. They possess a unique N-terminal

region comprising the before-mentioned poly-HAMP-domains (frequently more than 5) and, in contrast to different HHK of other groups, transmembrane regions cannot be found. HAMP domains are abundant in <u>h</u>istidine kinases, <u>a</u>denylyl cyclases, <u>m</u>ethyl-accepting <u>c</u>hemotaxis receptors and <u>p</u>hosphatases. They are supposed to act mainly as a kind of signal relay module transducing a variety of signals from sensor domains to output domains in these proteins. However, their function indeed might be more diverse. The domain structure of Amb07 is similar to the NIK1-kinase of *Candida albicans* involved in osmoregulation in the HOG pathway and susceptibility to fungicides like ambruticin VS-3 (Fig. S1)[15,18]. HAMP-deletions in NIK1 reduce the effectiveness of ambruticin[20]. Nevertheless, neither the mechanism behind these observations, nor the binding of the compound to NIK1 has been shown so far. In *M. xanthus* we identified two genes encoding such HHK, MXAN_0712 and MXAN_6735 (from here on referred to as AskA and AskB, respectively, for <u>a</u>mbruticin <u>s</u>ensor <u>k</u>inase A and B; Fig. S1). Interestingly, AskA has been reported to be essential for development[21]. Amb07, AskA, and AskB possess an extended N-terminal poly-HAMP region when compared to NIK1. We performed a de novo domain analysis of all proteins (NIK1, Amb07, AskA, and AskB) using SMART (Simple Modular Architecture Research Tool[22]). The N-terminal region was of particular interest (see Fig. S1). Besides a large number of predicted HAMP domains, putative partial SpoIIID-like domains comprising HTH_DeoR_1 domains (Pfam accession PF08220) could be found in Amb07, AskA, and AskB (9, 6, and 11, respectively) overlapping with the HAMP domains. Furthermore, putative partial HTH_6_RpiR domains (Pfam accession PF01418) also overlapping with the HAMP domains could be found in AskA and AskB (6 and 11, respectively). In AskA and AskB, these two partial domains always occurred in pairs with HTH_6_RpiR followed by HTH_DeoR_1. Such domains were not found in NIK1. Importantly, both types of domains usually occur in transcriptional regulators and mediate binding to DNA[23,24].

We generated and phenotypically characterized deletion mutants of the kinases (ΔaskA, ΔaskB, and the double mutant ΔaskAB). Either, FB formation in presence of ambruticin VS-3 and inhibition of FB formation in absence of ambruticin VS-3 in such mutants would indicate an involvement of the kinases in the process. However, no significant effects could be observed using either ΔaskA or ΔaskB under starvation with no additives (Fig. 2a). In contrast, the double deletion mutant ΔaskAB was unable to form FB under starvation conditions and in absence of glycerol or ambruticin VS-3 (Fig. 2a). Furthermore, ΔaskAB developed pseudospores in the absence of ambruticin VS-3 or glycerol. These pseudospores, besides being resistant to sonication, are able to germinate efficiently (Fig. 2b, c). The involvement of both kinases in sensing chemical inducers of spore formation was further shown by the phenotype of ΔaskA, ΔaskB, and ΔaskAB in attempts of induction of sporulation with ambruticin VS-3 and glycerol in liquid nutrient-rich CTT media. Interestingly, the ΔaskA and ΔaskB single mutants only transformed into spores in the presence of glycerol but were unable to perform chemically induced sporulation in the presence of ambruticin VS-3 (Fig. 3). In contrast, strain ΔaskAB was unable to react to either of the chemical inducers and no spores or "pseudospores" could be observed (Fig. 3). This result, together with the observed effect that the ΔaskAB mutant was unable to form FB under starvation conditions in the absence of additives, similar to the wild type DK1622 in the presence of ambruticin VS-3 or glycerol (Fig. 2a), led to the hypothesis that these proteins might be directly involved in sensing ambruticin VS-3. Consequently, we aimed to test direct binding of AskA and AskB to ambruticin VS-3 in vitro using heterologously expressed proteins by microscale

thermophoresis (MST) and surface plasmon resonance (SPR). However, using purified AskA and AskB, no binding to ambruticin VS-3 was observed, indicating the involvement of an additional factor (Table 1, Fig. S3). Nevertheless, AskA and AskB seem to interact tightly with each other as evidenced by a $K_D$ in the low micromolar range (Table 1, Figs. S2, S4).

Immediately downstream of *askB*, MXAN_6734 encodes for an additional HHK in *M. xanthus*, which we refer to as AskC (Fig. S1). Characterization of ΔaskC revealed an interesting phenotype. In contrast to wild type, ΔaskC was insensitive towards ambruticin VS-3 and glycerol under growth conditions (Fig. 3), but able to develop regularly even in presence of up to 60 nM ambruticin VS-3 or 300 mM glycerol under starvation conditions (Fig. 2a, c). These results show that AskC is essential for ambruticin VS-3 and glycerol sensitivity during starvation-induced development and during chemically induced sporulation.

AskC thus seemed to be involved in sensing ambruticin VS-3 directly or required for transduction of the signal. To elucidate this in depth, we aimed to test both, a direct interaction of AskC with the kinase AskA in vivo and a direct interaction of AskC with ambruticin VS-3 in vitro. A bacterial two-hybrid analysis was performed using different domains of AskA and AskC. An interaction between the receiver domain Rec3 of AskA and the receiver domain of AskC could indeed be observed (Fig. S5). In order to further elucidate the relation of AskA, AskB and AskC, purified AskA, AskB, and AskC was used in both, MST and SPR analyses (Table 1, Figs. S2, S3, S4). Besides the observed interaction of AskA and AskB, a specific interaction of both, AskA and AskB, with AskC exhibiting $K_D$'s in the low micromolar (AskA/AskC) or high nanomolar range (AskB/AskC) could be shown. Moreover, AskC was able to specifically bind to ambruticin VS-3 with a $K_D$ of 300 μM ± 200 μM. Suspected nonbinders exhibited no interactions (BSA, MXAN_0727 as proteins and myxovirescin as small molecule). However, the $K_D$ of the ambruticin VS-3-AskC interaction did not mirror an effective ambruticin VS-3 concentration of 4 nM in vivo. The results also raised the question if the kinases AskA, AskB and AskC form a three-membered complex or if only mutually exclusive pairs of kinases interact.

To complete the genetic analysis, double and triple mutants ΔaskBC, ΔaskAC, and ΔaskABC were constructed and phenotypically characterized (Figs. 2 and 3). Strain ΔaskBC was similar to strains ΔaskA or ΔaskB in phenotype as well as in response to ambruticin VS-3 and glycerol. This could be observed under both, starvation and growth conditions. Nevertheless, it has to be mentioned that ΔaskBC exhibited an unstable phenotype of FB formation under starvation in the absence of any inhibitor as FB were always formed but with a delay of up to 24 h (data not shown). The simultaneous inactivation of kinases AskA and AskC (ΔaskAC) resulted in a behavior similar to the wild type during starvation-induced development with no additives. Regarding FB formation, strain ΔaskAC was insensitive towards ambruticin VS-3 but not glycerol (Fig. 2a, b). However, the myxospores produced in these FB in presence of ambruticin VS-3 seem to germinate less efficiently (Fig. 2c). The chemical induction of sporulation under growth conditions was not possible using either inducer in this case. The phenotype of the triple mutant ΔaskABC resembled ΔaskAB under starvation. Nevertheless, this strain was still able to undergo chemically induced sporulation by the action of glycerol.

Taken together, the results of the genetic analysis indicated that either AskA or AskB has to be present in the cells to enable FB formation and sporulation under starvation. Only strains encoding AskA/AskC (ΔaskB), AskB/AskC (ΔaskA), AskA/AskB (ΔaskC), AskA (ΔaskBC) or AskB (ΔaskAC) are able to develop. AskC seems to be mainly responsible for sensitivity towards ambruticin VS-3 or glycerol under starvation, as indicated by the

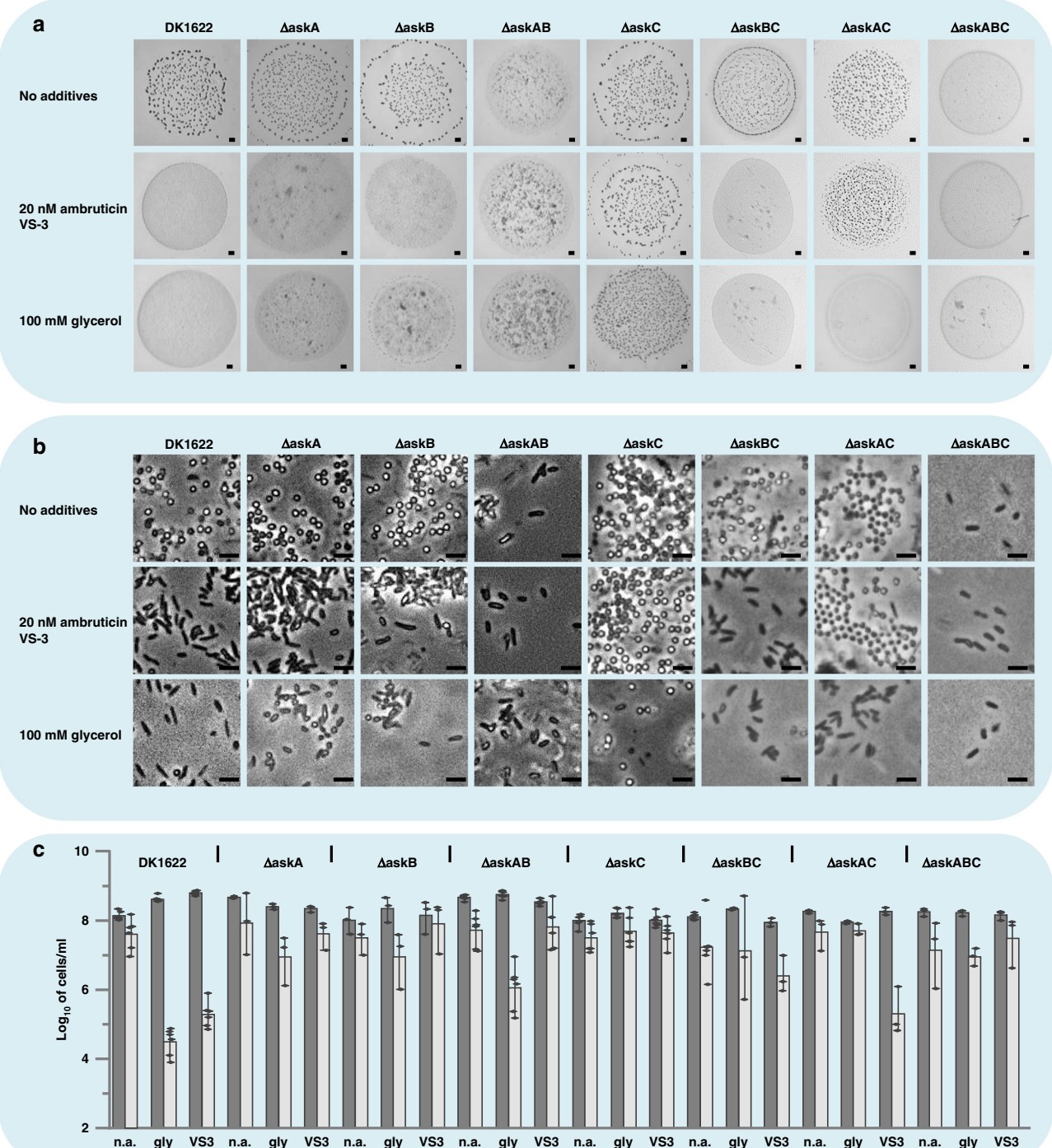

**Fig. 2 Phenotypical analysis of *M. xanthus* single-, double-, and triple *ask* mutants under starvation compared to DK1622. a** FB formation of *M. xanthus* single-, double-, and triple *ask* mutants on CF starvation medium without additives, or in the presence of 20 nM ambruticin VS-3 or 100 mM glycerol. Pictures were taken after 48 h of incubation. Scale bars represent 0.5 mm. **b** Spores and pseudospores of *M. xanthus* single-, double-, and triple *ask* mutants produced during FB formation on CF starvation agar in presence or absence of glycerol or ambruticin VS-3. Pictures were taken after 48 h of incubation. Scale bars represent 1 μm. The experiments were performed in triplicates. **c** Effect of 60 nM ambruticin VS-3 or 300 mM glycerol on differentiation of single-, double-, and triple *ask* mutants into sonication resistant cells (dark gray bars) and on germination efficiency (light gray bars) after 96 h of development. The order of the single attempts is marked below each pair of columns (n.a. no additives; gly glycerol; VS3 ambruticin VS-3). N = 3 or 6 biologically independent samples. The data are presented as mean values ± SEM. Source data are provided as a Source Data file.

behavior of the mutants ΔaskC and ΔaskAC. Nevertheless, the different responses of strains ΔaskC, ΔaskBC and ΔaskAC to ambruticin VS-3 and glycerol suggest that a balanced interplay of AskA and AskB is also required. In addition, the results suggest that ambruticin VS-3 inhibits the correct function of both, AskA

and AskB under participation of AskC. However, it is worth noting that in the absence of AskC and AskB, AskA is sufficient for a partial response to ambruticin VS-3 and glycerol, and in the absence of AskC and AskA, AskB is sufficient for a partial response to glycerol. Nevertheless, in both cases the pseudospores

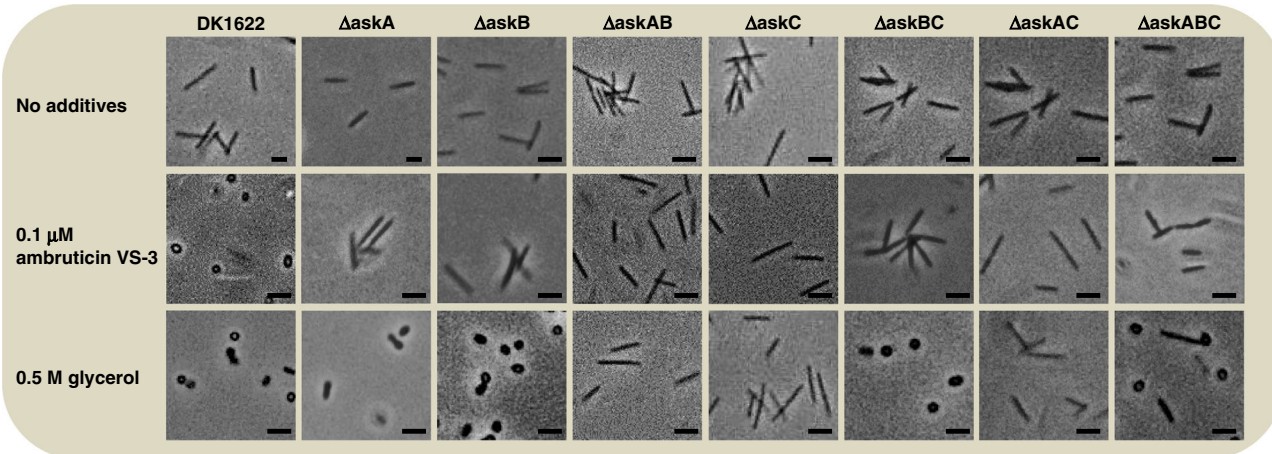

**Fig. 3 Chemical induction of sporulation of *M. xanthus* single-, double-, and triple *ask* mutants under growth conditions compared to DK1622.** Examples of chemically induced sporulation of *M. xanthus* single-, double-, and triple *ask* mutants compared to DK1622 wild type under nutrient-rich conditions by addition of 0.5 M glycerol and 0.1 μM ambruticin are shown. Pictures were taken 6 h after induction of sporulation. Scale bars represent 1 μm. The experiments were performed in triplicates.

**Table 1 Summary of tested AskA, AskB, AskC, MrpC, FruA, and ambruticin VS-3 interactions (including negative controls)[f].**

| tested interaction[e] | interaction[a] | $K_D$ (MST)[b] | $K_D$ (SPR)[b] | $K_D$ (EMSA)[c] | mean $K_D$[d] |
|---|---|---|---|---|---|
| AskA-AskB | y | 2.3 ± 0.1 | 3.0 ± 1 | n.d. | 2.8 ± 1.0 |
| AskA-AskC | y | 4.1 ± 0.1 | 1.4 ± 0.2 | n.d. | 2.8 ± 1.4 |
| AskB-AskC | y | 0.9 ± 0.4 | 0.4 ± 0.02 | n.d. | 0.7 ± 0.3 |
| AskA-MrpC | y | 0.5 ± 0.1 | 1.0 ± 0.04 | n.d. | 0.8 ± 0.3 |
| AskB-MrpC | y | 2.1 ± 0.9 | 0.3 ± 0.01 | n.d. | 1.2 ± 1.0 |
| AskC-MrpC | y | 1.4 ± 0.6 | 1.6 ± 0.05 | n.d. | 1.5 ± 1.0 |
| AskA-FruA | y | 1.4 ± 1.0 | 0.5 ± 0.06 | n.d. | 1.0 ± 1.0 |
| AskB-FruA | y | 2.1 ± 1.3 | 0.2 ± 0.03 | n.d. | 1.2 ± 1.0 |
| AskC-FruA | y | 1.3 ± 1.0 | 1.5 ± 0.2 | n.d. | 1.4 ± 1.0 |
| AskA-MXAN_0727 | n | — | — | n.d. | — |
| AskB-MXAN_0727 | n | — | — | n.d. | — |
| AskC-MXAN_0727 | n | — | — | n.d. | — |
| AskA-BSA | n | — | — | n.d. | — |
| AskB-BSA | n | — | — | n.d. | — |
| AskC-BSA | n | — | — | n.d. | — |
| AskA- ambVS3 | n | — | — | n.d. | — |
| AskB- ambVS3 | n | — | — | n.d. | — |
| AskC- ambVS3 | y | 159 ± 19 | 512 ± 128 | n.d. | 300 ± 200 |
| AskC-myvA | n | — | — | n.d. | — |
| MrpC-ambVS-3-P$_{mrpC}$ | y | n.d. | n.d. | <0.005 | — |
| MrpC-ambVS-3 | n | — | — | n.d. | — |

[a] y = interaction, n = no interaction observed.
[b] $K_D$ [μM] based on triplicate experiments.
[c] estimated $K_D$ [μM] based on EMSA.
[d] average $K_D$ [μM] calculated based on $K_D$ values obtained from MST and SPR.
[e] ambVS3 = ambruticin VS-3, myvA = myxovirescin A.
[f] dose response curves of all MST analyses and Rmax-plots of SPR analyses revealing protein interactions can be seen in Figs. S2, S3, S4; P$_{mrpC}$ = DNA bearing promoter of *mrpC*; n.d. = not determined.

formed by the double mutants germinate more efficiently than those formed by DK1622. The chemical induction of spores by both, ambruticin VS-3 and glycerol is also regulated by the interplay of the three kinases. However, both chemical inducers might not share the identical downstream pathway, which is indicated by the striking difference in the response of the mutant strains to either ambruticin VS-3 or glycerol under growth conditions: AskA, AskB and AskC have to be functional to enable ambruticin VS-3 induced chemical sporulation under growth conditions. In contrast, this is not necessary for glycerol-induced sporulation.

**The ambruticin signaling pathway enters the developmental GRN upstream of MrpC/FruA.** An overview of the

developmental GRN of *M. xanthus* has been provided by Kroos in 2017[8] (see below for the central mrp-module of the GRN). In order to elucidate how AskA, AskB and AskC interfere with the known components of the GRN, a series of promoter-*lacZ* fusions of developmental genes were used to narrow down the entry point. Those included the key regulator genes *mrpC* and *fruA*, the *dev* operon and the *exo* operon. MrpC and FruA are regulators of hundreds of developmental genes, also including the *dev*- and *exo* operons[1,8,10,25–30]. The *dev* operon is located directly upstream of the *exo* operon in the hierarchy where the *exo* operon is involved in both, developmental and chemically induced sporulation[1,8,13,31]. All promoters (P) were tested in ΔaskAB or ΔaskC, in absence and presence of ambruticin VS-3 or glycerol and compared to wild type activities (Fig. S6a and S6b). Addition

of both, ambruticin VS-3 and glycerol led to significantly reduced activities of $P_{mrpC}$ and $P_{fruA}$ in the wild type. The presence of ambruticin VS-3 did not affect those promoter activities in $\Delta$askC, whereas in $\Delta$askAB, both promoters were downregulated without addition of the effectors to a similar extent as observed in the wild type upon addition. This indicated that AskA, AskB and AskC interfere with the GRN at or upstream the MrpC/ FruA level. The response pattern of $P_{dev}$ was identical to the one observed for $P_{mrpC}$ and $P_{fruA}$. In contrast, and surprisingly, $P_{exo}$ was induced upon addition of ambruticin VS-3 or glycerol. This was not visible or less pronounced in both, $\Delta$askAB and $\Delta$askC mutants, for ambruticin VS-3 and/or glycerol.

The increasing activity of promoter $P_{mrpC}$ early during development is a prerequisite for successful FB formation under starvation conditions[8]. Due to the abovementioned effects on the activity of $P_{mrpC}$ in strains DK1622, $\Delta$askAB and $\Delta$askC, we concluded that $P_{mrpC}$ might be the most-likely entry point of the ambruticin VS-3 or glycerol-induced signal to the GRN. However, the involvement of the three interacting group III HHK suggests the presence of a complex phosphorelay system which might influence the GRN at multiple points. Nevertheless, $P_{mrpC}$ is early in the hierarchy of the GRN. Due to the obvious influence of the kinases, we focused on the activity of $P_{mrpC}$ under starvation in all additional mutants (Fig. S6c). In general, a significant increase of the activity of $P_{mrpC}$ co-occurred with an observed formation of FB and sporulation under starvation (compare to Fig. 2a). A decreased activity co-occurred with a lack of FB formation and led to the occurrence of pseudospores. Two exceptions could be observed: In strain $\Delta$askBC, $P_{mrpC}$ activity first increased at the onset of development to wild type levels (up to 18 h post induction of starvation) followed by a steep decrease until 24 h. This was independent of the presence of both, ambruticin VS-3 and glycerol. As mentioned before $\Delta$askBC is indeed able to develop but only in the absence of ambruticin VS-3 and glycerol. Also, the strain exhibited an unstable delay of approximately 24 h regarding FB formation. However, the remaining activity of $P_{mrpC}$ in the absence of ambruticin VS-3 or glycerol seems to be sufficient to allow further development. It is tempting to consider this finding as an indication for a second entry point of the kinases into the GRN of development. In strain $\Delta$askAC, $P_{mrpC}$ activity increases independently of the presence of ambruticin VS-3 or glycerol to a high level. Although $\Delta$askAC is not affected by ambruticin VS-3, its FB formation is abolished by the presence of glycerol under starvation. This might be another hint for a second entry point of the kinases to the GRN. Although the kinases per se are involved in response to both, ambruticin VS-3 and glycerol, these activities of $P_{mrpC}$ not matching to the phenotype are an indication that the signals induced by both compounds do not share or activate identical downstream pathways.

**Ambruticin affects MrpC binding to its own promoter.** MrpC is one key component in the starvation- and glycerol-induced sporulation process[10,13,32]. We analyzed the activity of all reporter promoters in a $\Delta$mrpC mutant. For all promoters tested, no difference in activity was observed under developmental conditions when ambruticin VS-3 or glycerol was added to the medium using $\Delta$mrpC (Fig. S6a). However, in $\Delta$mrpC, $P_{mrpC}$ and $P_{exo}$ were constitutively upregulated but also not reacting to any inducer. In the case of $P_{mrpC}$ this suggests that upon starvation, the early developmental pathway seems to be unaffected by ambruticin VS-3 and glycerol. This also includes phosphorylation of MrpB, which activates *mrpC* transcription. Constitutive upregulation of $P_{mrpC}$ in a $\Delta$mrpC background suggests an auto-inhibitory effect on the promoter which has been reported

previously[32]. The constructed $\Delta$mrpC mutant also failed to sporulate upon addition of ambruticin VS-3 in nutrient-rich CTT medium (not shown), again highlighting the essential function of MrpC in chemically induced sporulation.

To elucidate if ambruticin VS-3 affects binding of MrpC to its own promoter in vivo, we performed a pull-down experiment using a biotinylated DNA fragment comprising $P_{mrpC}$. As the focus of our work was more on the role of ambruticin VS-3 in the observed interspecies interaction, glycerol was not included in this experiment. The bait was incubated with proteinaceous extracts of developmental cells grown in the presence or absence of ambruticin VS-3. Indeed, MrpC bound to $P_{mrpC}$, which was found only when using a cell extract previously exposed to ambruticin VS-3 (Fig. 4a). Such an alteration of MrpC binding affinity to $P_{mrpC}$ could provide an explanation for why ambruticin VS-3 leads to an inhibition of development. Although AskC was able to bind to ambruticin VS-3 (see above), the observed $K_D$ of the ambruticin VS-3-AskC interaction did not mirror an effective ambruticin VS-3 concentration of 4 nM in vivo. Therefore, and as we could observe a significant difference in binding of MrpC in presence of ambruticin VS-3 in the pull-down experiment, we performed an EMSA assay using purified MrpC to monitor binding of MrpC to $P_{mrpC}$ in presence of ambruticin VS-3. Surprisingly, MrpC binding to $P_{mrpC}$ seemed to be strongly altered already at 5 nM ambruticin VS-3, the lowest concentration used in the EMSA assay, suggesting that MrpC forms a strong complex with DNA in presence of ambruticin VS-3 and thereby explaining the low effective concentration of ambruticin VS-3 in vivo (Fig. 4b).

**MrpC and FruA bind to AskA, AskB and AskC in vitro.** EMSA assays revealed an altered DNA-binding behavior of MrpC to $P_{mrpC}$ in the presence of ambruticin VS-3, thereby possibly explaining a repression of developmental genes caused by ambruticin VS-3. In contrast, AskA, AskB and AskC are obviously involved in ambruticin-sensing, where AskC specifically binds to ambruticin VS-3. In an attempt to explain this, we tested whether AskA, AskB or AskC are able to interact with MrpC (Table 1, Figs. S2, S4). Indeed, each of the kinases alone is able to interact specifically with MrpC exhibiting $K_D$ values in the high nanomolar (AskA-MrpC) or in low micromolar range (AskB-MrpC and AskC-MrpC). Interestingly, FruA also specifically interacted with all three kinases. Again, $K_D$-values were in the low micromolar range. All other proteins used as negative controls exhibited no binding to MrpC or any of the kinases (BSA and MXAN_0727). The interaction of MrpC and ambruticin VS-3 alone was also assayed by SPR, but no clear interaction in the absence of DNA was observed (not shown), indicating the need for DNA as a part of a stable ambruticin VS-3-DNA-MrpC complex (Fig. 4a, b).

We conclude that, both key regulators of development, MrpC and FruA, are able to undergo specific pairwise interactions with the kinases, a fact possibly explaining how a second entry point in the GRN of development might be realized. These interactions of both key regulators with the kinases are also interesting as this is an indication that MrpC and/or FruA might be phosphorylated by the kinases in vivo. It has been shown that MrpC is phosphorylated at its N-terminus by the serine/threonine protein kinase (STPK) Pkn14 in a STPK cascade which consists of kinases Pkn8/Pkn14[33–35]. Recently, a cluster of threonines and serines (TTSS motif) in the amino-terminal region was identified to be essential for MrpC activity in vivo[36]. The motif was not phosphorylated by Pkn14, but was necessary for Pkn14-mediated phosphorylation of other sites in MrpC. However, Pkn8 and Pkn14 seem not to be active during development in *M. xanthus.*

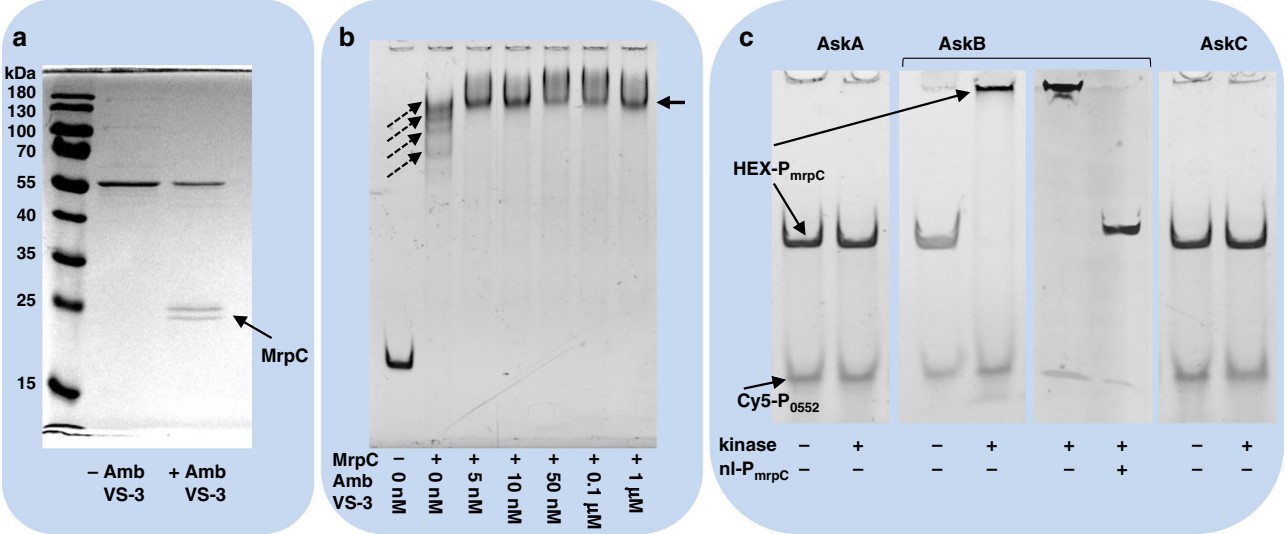

**Fig. 4 Interaction of MrpC and kinases AskA, AskB and AskC at the promoter region of *mrpC*. a** MrpC can be isolated from a proteinaceous lysate of cells preincubated with ambruticin VS-3 using PmrpC-bearing DNA (PmrpC-DNA). A representative photograph of the samples isolated in a DNA-protein pull down assay after SDS polyacrylamide gel electrophoresis (SDS-PAGE) is shown. Left: PageRuler prestained protein ladder (Thermo). All bands were excised and proteins were identified by MALDI-Tof analysis. MrpC, two bands marked by an arrow. The prominent band at 55 kDa in both attempts was identified to be a peroxiredoxin-like protein (AhpC; MXAN1564). **b** Influence of ambruticin VS-3 on binding of MrpC to PmrpC-DNA. A scan of a representative native PAGE of one attempt shows the signals of HEX-labeled PmrpC-DNA. Addition of MrpC leads to four shifted bands of PmrpC-DNA indicating various binding sites (dashed arrows). Upon addition of ambruticin VS-3, only one shifted band is visible (arrow). In both, presence and absence of ambruticin VS-3 the shift of PmrpC-DNA is quantitative when compared to the attempt without MrpC or ambruticin VS-3 (lane 1) indicating a different arrangement of MrpC at the DNA because of ambruticin VS-3 action. **c** Interaction of AskB with PmrpC-DNA as determined by EMSA. A scan of a representative native PAGE gel of one attempt of a competition assay is depicted showing the signals of HEX-labeled PmrpC-DNA (HEX-$P_{mrpC}$) as well as Cy5-labeled Pmxan_0552-DNA (Cy5-$P_{0552}$). The addition of each kinase AskA, AskB and AskC is indicated below each lane. Equal amounts of labeled PmrpC-DNA and labeled Pmxan_0552-DNA were used. Bands of each type of DNA are marked. The addition of non-labeled PmrpC-DNA (nl-$P_{mrpC}$) is indicated below each lane. Results shown in panels **b** and **c** were performed in triplicates. Results shown in panel **a** were performed in duplicates. Source data are provided as a Source Data file.

AskA, AskB and AskC would be good candidates for such active kinases under starvation as their involvement in development has now been shown. FruA has been proposed to be phosphorylated but no cognate kinase has been identified so far and various observations indicate that no phosphorylation event is involved in modulation of FruA activity[9,29,37]. However, the key finding is that two proteins affected by ambruticin VS-3 (AskC as direct binder and MrpC exhibiting altered binding of to $P_{mrpC}$ in presence of ambruticin VS-3) are able to interact specifically.

**The kinase AskB specifically binds to $P_{mrpC}$ in vitro**. To further elucidate the relationship between AskA, AskB, AskC and MrpC, we performed preliminary EMSA assays that also included the kinases. A remarkable effect was observed when $P_{mrpC}$-bearing DNA ($P_{mrpC}$-DNA) was incubated with AskB (Fig. 4c). AskB exhibited specific binding to the promoter of $P_{mrpC}$ as could be concluded by a clear shift of only $P_{mrpC}$-DNA (fluorescently labeled) in a competition assay in the presence of unrelated promoter DNA ($P_{mxan\_0552}$). Obviously, a large complex was formed, as the $P_{mrpC}$-DNA was shifted to the well of the gel. This could not be observed in the case of AskA or AskC. To further confirm this specificity, we introduced a large excess of non-labeled $P_{mrpC}$-DNA. At this, the observed binding to the labeled $P_{mrpC}$-DNA was abolished thereby confirming a specific binding of AskB to $P_{mrpC}$ in vitro. To our knowledge, such DNA-binding ability of a HHK is unprecedented. However, additional experiments in vivo have to be performed to further elucidate this exceptional finding. Nevertheless, this observation, together with our other results suggests the existence of one or more complexes containing AskA, AskB, and/or AskC able to interact

with both MrpC and FruA, and $P_{mrpC}$-DNA, therefore likely to be key developmental regulators. AskB (in vitro) and MrpC (in vivo) are both able to specifically bind to and most probably in a concerted action suppress $P_{mrpC}$ transcription under non-starving conditions.

**AskAB and AskA bind to $P_{mrpC}$ in vitro**. The pairwise interactions of each kinase and MrpC, together with the evidence in vitro for binding of AskB to $P_{mrpC}$ raised the question whether a complex of all proteins may exist or only mutually exclusive pairs are possible. In an attempt to investigate this question, we performed pull-down approaches either using biotinylated $P_{mrpC}$-DNA (DNA-protein pull-down) or C-terminally-His-tagged AskC (AskC-His; protein-protein pull-down) as a probe. The different probes were used to extract single proteins or protein complexes from mixtures with different composition of AskA, AskB, AskC, MrpC and ambruticin VS-3. Proteins able to bind to $P_{mrpC}$ or AskC-His were extracted from the mixtures using biotinylated $P_{mrpC}$-DNA attached to streptavidin-coated magnetic beads and AskC-His attached to $Co2^{+}$-Dynabeads, respectively. As a negative control, either streptavidin-coated beads without $P_{mrpC}$-DNA or Dynabeads without AskC-His were used. Proteins able to bind to either $P_{mrpC}$-DNA or AskC-His in vitro were analyzed by SDS-PAGE. In the protein-protein pull-down attempts, we could confirm the pairwise interactions of AskC with AskA or AskB as already seen using MST and SPR analysis (Fig. 5b). Both kinases AskA and AskB could also be extracted together in one attempt when using AskC-His as a probe. However, this observation did not necessarily account for the presence of a three-membered complex containing AskA, AskB,

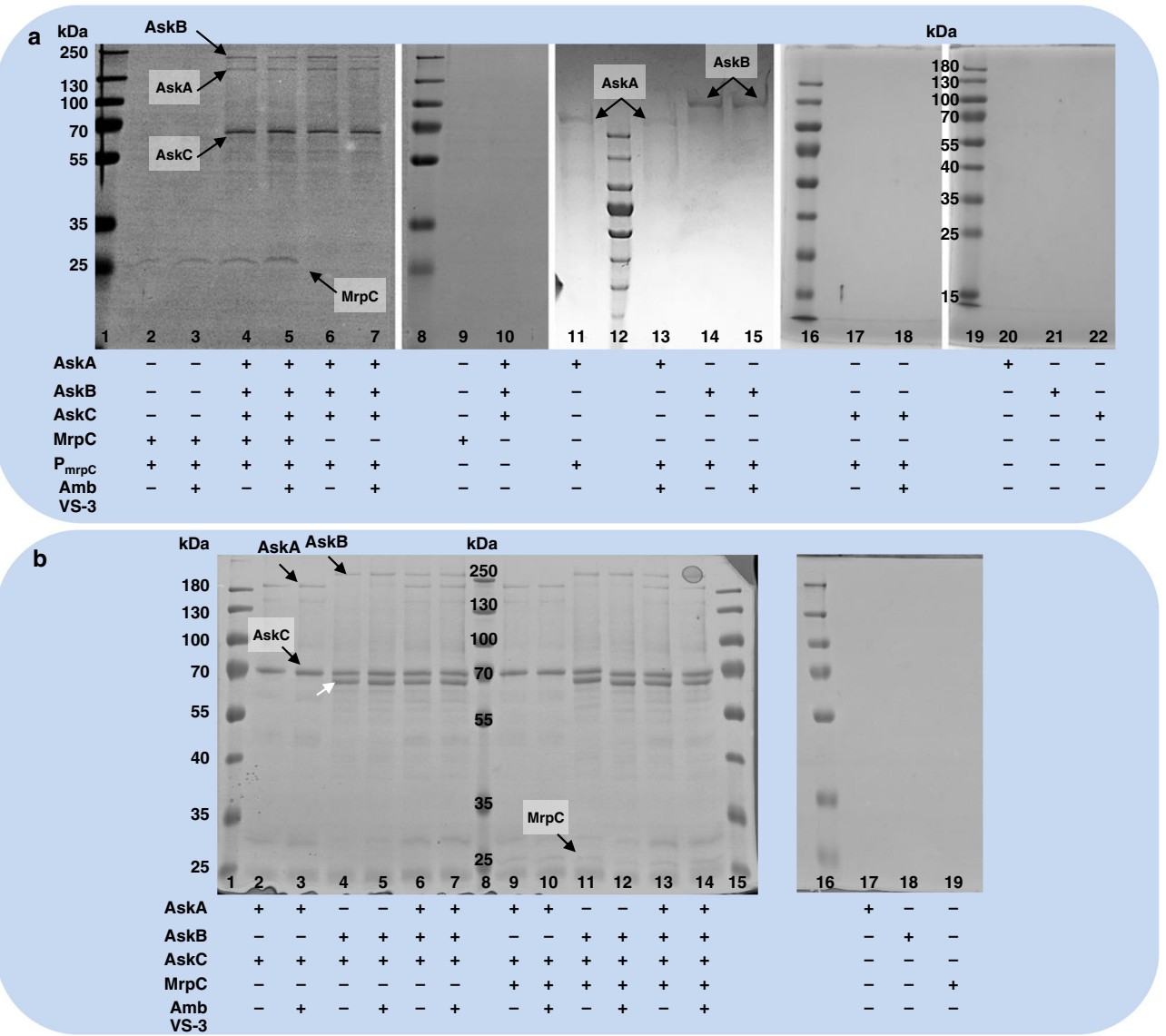

**Fig. 5 Interaction of MrpC and kinase complexes at the promoter region of *mrpC*. a** Interaction of AskA, AskB, AskC and MrpC with PmrpC-DNA as determined by DNA-protein pull-down assays. Scans of representative SDS-PAGE gels resolving proteins captured in various DNA-protein pull-down experiments are shown. Protein bands are marked and named exemplarily. Biotinylated PmrpC-DNA (probe), ambruticin VS-3, as well as purified proteins AskA, AskB, AskC, and MrpC were mixed in different combinations in the presence of ATP. Proteins, which bound to the biotinylated PmrpC-DNA, were extracted from the mixtures by using streptavidin-coated magnetic beads. The composition of each attempt is indicated below the lanes. Lanes 1 and 8: PageRuler plus prestained protein ladder (Thermo); Lanes 12, 16, and 19: PageRuler prestained protein ladder (Thermo). **b** Interaction of AskA, AskB, AskC and MrpC as determined by protein-protein pull-down assays. Scans of representative SDS-PAGE gels resolving proteins captured in various protein-protein pull-down experiments are shown. Protein bands are marked and named exemplarily. The protein band marked by a white arrow is an N-terminal fragment of AskB, which could not be separated from full-length AskB upon purification. Other minor impurities are all smaller N-terminal fragments of AskA or AskB, which also could not be separated. C-terminally His6-tagged AskC (probe), ambruticin VS-3, as well as purified proteins AskA, AskB and MrpC were mixed in different combinations. Proteins, which bound to the C-terminally-His's-tagged AskC, were extracted from the mixtures by using Dynabeads® His-Tag Isolation & Pulldown magnetic beads. The composition of each attempt is indicated below the lanes. Lanes 1 and 15: PageRuler prestained protein ladder (Thermo); Lanes 8 and 16: PageRuler plus prestained protein ladder (Thermo). Results shown in panels **a** and **b** were performed in duplicates. Source data are provided as a Source Data file.

and AskC. Addition of ambruticin VS-3 did not cause any change in the interactions. In the DNA-protein pull-down assays, we could again observe interaction of AskB alone with $P_{mrpC}$-DNA (Fig. 5a), supporting the result of specific binding of AskB to $P_{mrpC}$-DNA in EMSA assays. In addition to this, AskA also exhibited such interaction with $P_{mrpC}$-DNA. The latter finding was unexpected, as we could not see interaction of AskA and biotinylated $P_{mrpC}$-DNA before in EMSA assays (Fig. 4c). As mentioned above, at least AskA or AskB is needed for

development (Fig. 2). In AskA and AskB, a large number of partial and pairwise occurring putative HTH_6_RpiR domains and putative HTH_DeoR_1 domains could be found (Fig. S1). An interaction of AskA with DNA would consequently make sense but still has to be proven in vivo. However, both proteins are able to replace each other at least partly in function. Furthermore, AskA, AskB, and AskC could be isolated using $P_{mrpC}$-DNA indicating that at least a two-membered complex of kinases (either AskB/AskC or AskA/AskC or both) is located at

$P_{mrpC}$-DNA in vitro. This was independent of the presence of MrpC and the binding of MrpC and the kinases seemed not to be mutually exclusive. Again, ambruticin VS-3 did not influence the composition of proteins isolated in the assays.

All the observations relate to the in vitro setting and need to be analyzed further in vivo. Nevertheless, our data provide compelling evidence for DNA binding and will enable future research to unravel the overall mechanism of action of AskA, AskB, and AskC together with the main regulators of development in *M. xanthus*.

## Discussion

Development of *M. xanthus* requires a complex GRN involving hundreds of genes[2,3,8]. Although thoroughly investigated, many questions remain unanswered. Here, we studied interactions of *M. xanthus* with co-inhabitants of their environment to reveal hidden aspects of the GRN. This enabled the identification of ambruticin VS-3 as an inhibitor of starvation-induced development and as an inducer of sporulation during growth. Experiments were also performed using glycerol, a known chemical inducer of spore formation. Similar effects on both, regulation and phenotype were visible, suggesting comparable roles of chemical inducers of sporulation. Under nutrient-rich conditions, ambruticin VS-3 induces sporulation. The stimulus triggering chemically induced sporulation in nature remained elusive for decades, as all inducers tested previously require concentrations unlikely to occur in natural habitats[1,11,13,31,32]. Being at least 1.000.000 times more potent than glycerol, and produced at effective concentrations by species commonly present in soil habitats, we can designate ambruticin VS-3 as a natural inducer of chemically induced sporulation. Nevertheless, the potency of ambruticin VS-3 in comparison to all other known inducers mentioned above might also be interpreted in a slightly different way but leading to a more comprehensive view. Ambruticin VS-3 produced by *S. cellulosum* So ce10 obviously enables the producer to force *M. xanthus* into sporulation albeit nutrients are present. The consequence is that *M. xanthus* is not able to propagate in the ecological niche and slower growing species like the ambruticin producer are able to make use of this obvious advantage. The consequence would be that ambruticin VS-3 is rather a distinct and powerful tool of a producer to mislead competing species by handing over "false information" about availability of nutrients. As indicated by previous work, the chemically induced sporulation might also be considered as an "emergency program" of vegetative cells faced with adverse conditions although the presence of nutrients would support vegetative growth. This is indicated by work of O'Connor and Zusman, which shows that agents causing damage to the peptidoglycan induce sporulation[12,38]. The presence of glycerol in high amounts mimics another adverse condition well-known as hyperosmotic stress. Consequently, an immediate reaction of vegetative cells is necessary. Under starvation conditions, ambruticin VS-3 inhibits FB formation and affects sporulation, inducing a differentiation into cells with properties of vegetative cells and myxospores, which we refer to as "pseudospores". Although pseudospores produced by the wild type in the presence of ambruticin VS-3 resemble those of myxospores, a significant decrease of germination efficiency and a difference in morphology is obvious. Again, it seems that So ce10 derives a relevant benefit from the situation by the action of ambruticin VS-3. *M. xanthus* is not able to aggregate and to perform regular spore formation resulting in a clear disadvantage for *M. xanthus* in trying to survive a threatening shortage of nutrients. However, to enable a more comprehensive overview of the ecological relevance of these observations, additional pairings would have to be examined in co-cultivation. Nevertheless, we could show that ambruticin VS-3 is efficiently produced by So ce10 under these conditions and reaches effective levels already 24 h post inoculation. A first insight into a spectrum of potential target myxobacteria could be obtained when searching for myxobacteria encoding close homologs of all three kinases AskA, AskB and AskC. Amongst all published myxobacterial genomes, homologs of AskA, AskB and AskC could be found almost exclusively in the suborder Cystobacterineae (data not shown) which *S. cellulosum* is not part of. The kinase genes seemed to be highly conserved (query coverage above 99% with a pairwise identity of 63.3 to 100%) indicating that ambruticin VS-3 may be effective against various if not all members of the respective suborder.

AskA (MXAN_0712) has been shown to be essential for development[21]. However, in our hands neither constructed in frame deletions of *askA* nor of *askB* exhibited a developmental defect. Such phenotype could only be observed in either a ΔaskAB or a ΔaskABC background. As our entire in frame deletions were verified by both, PCR and sequencing, we hypothesize that the most likely reason for this discrepancy might be a different genetic background in both variants of *M. xanthus* DK1622[39]. To elucidate this, genome sequencing and comparison of both genomes have to be performed.

Based on our work we could identify the three kinases AskA, AskB, and AskC as part of a central switch of both, starvation-induced development and chemically induced sporulation. The analysis of all constructed mutants revealed that either AskA or AskB has to be functional to enable development under starvation conditions. AskA and AskB are obviously able to substitute for each other in function under these conditions. Nevertheless, a different response of some of these mutants after addition of ambruticin VS-3 or glycerol could be observed. The importance of AskC as direct binder of ambruticin VS-3 and its role in sensitivity towards both, ambruticin VS-3 or glycerol could be shown. However, based on the phenotypical analysis and the examined promoter activities of $P_{mrpC}$, we reason that ambruticin VS-3 finally seems to inhibit the correct function of AskA or AskB by its action at AskC during starvation-induced development. In contrast, any inactivation of either AskA, AskB or AskC led to a striking inability of the mutants to undergo chemically induced sporulation through the action of ambruticin VS-3. This was not true in the case of glycerol-induced sporulation. Strains still encoding either *askB/askC*, *askA/askC* or *askA* only could be forced to sporulate upon the addition of glycerol. However, strain ΔaskABC encoding none of the kinases was also able to respond to glycerol. Based on the abovementioned data, we hypothesize that on the one hand, AskA, AskB and AskC influences the GRN at additional entry point(s). This might be reflected by the specific interaction of each kinase with FruA, the second main regulator of development, as well as the high number of different receiver domains in the kinases. On the other hand, we revealed that ambruticin VS-3 acts as a direct binder and show that this is rather unlikely for glycerol as the mutants react in such a different way compared to the addition of ambruticin VS-3. Although we could show similar phenotypes and effects on promoters upon addition of glycerol and ambruticin VS-3, it remains elusive whether and how glycerol is interacting with intracellular AskA, AskB or AskC. Hyperosmotic extracellular conditions due to the presence of extracellular glycerol might be sensed by receptor-type adenylyl cyclases CyaA and CyaB described as involved in osmosensing in *M. xanthus*[40,41]. This signal would have to be transduced to the Ask kinases. Further candidates for an interaction with the Ask kinases could be osmosensing transmembrane receptors like group VI HHK Sln1 of *Saccharomyces cerevisiae* which is the osmolarity sensor of the HOG pathway affected by ambruticin in fungi as summarized recently[19]. Nevertheless, we

could not identify similar proteins in *M. xanthus*. However, altering the intracellular level of glycerol has been reported to play some role in development[42]. Consequently, it might be possible that high extracellular amounts of glycerol lead to increased concentrations of glycerol in the cytosol, thereby interacting directly with the Ask kinases or MrpC and triggering an effect like ambruticin VS-3 via a similar molecular mechanism. This might also explain the high concentrations of glycerol, which are necessary to trigger an ambruticin-like effect, assuming glycerol is not actively imported into the cell.

The underlying molecular mechanisms clearly demand further experimental work to be understood in more detail. However, we made a number of striking observations using purified proteins AskA, AskB, AskC, MrpC, and FruA in vitro and are therefore able to present insights into their complex interaction. Based on these findings we propose that complexes involving two or three kinases and MrpC directly regulating $P_{mrpC}$ play a key role in both, regulation of starvation-induced development and chemically induced sporulation in the so-called mrp-module (as reviewed by Kroos[8]) (Fig. 6). In vitro, a complex consisting of either AskA/AskC, AskB/AskC, AskA/AskB, or AskA/AskB/AskC is possible. In each combination, the resulting complex is able to interact with MrpC and/or with $P_{mrpC}$-DNA. No visible effect on these interactions was observed upon addition of ambruticin VS-3 in the respective pull-down experiments. Therefore, the effect of ambruticin VS-3 on the interplay of these proteins, and consequently on the final output, might be related to phosphorylation events, the basic function of a kinase. Another explanation might be a lack of additional essential players in vitro again underpinning the need for additional in vivo data. Such lacking factors could be additional proteins or cofactors including intracellular starvation signals, which were not included in the experiments. These starvation signals obviously are of importance, as the phenotypes of the kinase mutants under starvation differ significantly compared to growth conditions upon addition of the chemical inducers. Nevertheless, regarding the hierarchy of the known GRN of development, the GRN is influenced at $P_{mrpC}$ by the concerted action of AskA/AskB/AskC and MrpC as demonstrated by our examinations. The DNA-binding ability of kinase AskB was shown in vitro by two independent approaches. Specificity of AskB for binding to $P_{mrpC}$-DNA could be demonstrated. In both kinases AskB and AskA, a series of putative partial but pairwise occurring HTH_6_RpiR- and HTH_DeoR_1-like DNA-binding domains could be identified in silico in the N-terminal regions. No such domains could be found in AskC or the fungal group III HHK Nik1. Intriguingly, AskA also exhibited binding to $P_{mrpC}$ in vitro in a pull-down approach. These findings are of potential broad interest regarding the DNA-binding affinity of kinases and require further analyses in vivo.

There is evidence for a bipartite action of ambruticin VS-3 in the interplay of AskA, AskB, AskC and MrpC. First, AskC has been shown to bind specifically to ambruticin VS-3 in SPR and MST analysis. Second, ambruticin VS-3 significantly alters the binding behavior of MrpC to its own promoter as shown in EMSA analysis. However, using SPR, no binding of MrpC and ambruticin VS-3 could be observed indicating that a complex of both, MrpC and DNA comprising the $P_{mrpC}$ promoter is needed to enable ambruticin VS-3 interaction. Similar intriguing interactions have been reported for other secondary metabolites which were shown to address more than one target in parallel like the disorazols[43].

Although we are still far from a comprehensive understanding of the entire molecular mechanism, we propose a revision of the mrp-module of development where interacting pairs of the kinases or even a complex AskA/AskB/AskC in an interplay with both, MrpC and/or FruA act as a main switch for development

like described in Fig. 6b. This might also include some sequestration of MrpC or FruA as has recently been suggested for FruA[37]. Similar complexes may bind to additional promoters and might involve FruA. However, this hypothesis clearly demands further experimental work. Striking questions of the molecular mechanism have to be addressed. They include the issue whether MrpC and/or FruA are permanently associated with the kinases or under which conditions dissociation might occur. Whether both regulators interact with kinase complexes in parallel and thereby regulate promoters including $P_{mrpC}$ currently also remains elusive. However, the observations described herein will enable redirected research to understand how MrpC and FruA are able to tightly coordinate the regulation of hundreds of genes during development.

Our results also allow for a different view of the fungal target of ambruticin VS-3: Although it is known that a group III HHK is essential for susceptibility, a direct interaction has yet to be proven[20,44]. It is tempting to speculate that a complex of fungal HHKs has to interact with an additional transcriptional regulator triggering susceptibility via upregulation of the HOG pathway.

The findings about the interplay of AskA, AskB, AskC, MrpC and FruA reported herein will significantly influence the understanding of developmental regulation in myxobacteria. A new perspective on the fungal target of ambruticin VS-3 and our in vitro evidence for AskB as a DNA-binding HHK will also influence research directions in other organisms.

## Methods

**Bacterial strains, plasmids, growth conditions, and nucleic acid manipulations.** All bacterial strains, plasmids, and oligonucleotides used in this study are listed in Tables S1, S2, and S3, respectively. *M. xanthus* strains were grown in CTT broth or CTT agar plates (1.5% Bacto-agar, Difco)[45] at 30 °C. For starving conditions, CF agar plates[45] were used. Mid-log grown cells were adjusted to an $OD_{600}$ of 7.5 ($2.25 \times 10^9$ cells/ml) in liquid CF medium. The concentrated cells were spotted on CF plates and incubated at 30 °C. For calculation of sonication resistance and germination efficiency, $1.35 \times 10^9$ cells were left to develop during 96 h at 30 °C. They were harvested and resuspended in 2 ml of liquid CF. Several dilutions of the samples were subjected to sonication for number of sonication resistance cells, or to heat treatment (1 h incubation at 50 °C) followed by sonication for germination. The number of sonication resistance cells was determined using a Neubauer counting chamber and the germination efficiency by plating the treated dilutions on CTT plates and counting the colonies growing on the plate after 5 days of incubation at 30 °C. When needed, media was supplemented with kanamycin (80 μg/ml), sucrose (50 mg/ml), or different concentrations of ambruticin (ambruticin VS3[14]) or glycerol. Four nM ambruticin VS-3 is sufficient to induce pseudospores, but, as can be seen in the fruiting bodies picture (4 nM in Fig. 1b), the phenotype is not completely expressed (mounds are still visible indicating an initial aggregation of cells). Consequently, we used 20 nM ambruticin VS-3 for further qualitative analysis, e.g., for pictures of development, as this is triggering a full response which can be seen in Fig. 1b. As the optimal glycerol concentration required for chemically induced sporulation on rich medium is 5 times the amount required for fruiting body inhibition on plate, we used the same ratio for induction with ambruticin VS-3 (i.e., 100 nM). To avoid experimental errors derived from effector-concentrations too close to the minimal requirements, we used concentrations that were 3 times the minimal requirements (i.e., 60 nM ambruticin VS-3 and 300 mM glycerol) in the quantitative analysis such as sporulation or during growth of the strains bearing promoter-*lacZ* fusions. For the co-incubation experiments of different myxobacteria VY2-agar supplemented with 2 g/L starch was used[46]. *E. coli* strains were grown on LB broth or LB agar plates (1.5% agar)[47] at 37 °C. When necessary, kanamycin (50 μg/ml), ampicillin (100 μg/ml), and/or isopropyl-β-D-thiogalactopyranoside (IPTG, 0.5 mM) was added. For bacterial two-hybrid analysis, *E. coli* BTH101 was plated on M63 agar plates supplemented with 25 μg/ml kanamycin, 50 μg/ml ampicillin, 0.5 mM IPTG and 40 μg/ml X-Gal. For genetic manipulations, routine molecular biology techniques were used[47]. Plasmids were introduced into *E. coli* and *M. xanthus* by electroporation[48].

**Extraction of ambruticin VS-3 from agar plates.** For the determination of the ambruticin VS-3 content in VY2-agar during the growth of strain So ce10, cells of So ce10 grown in VY2 medium were adjusted to a cell density of $10^{10}$ cells per ml. A total of 100 μl of the suspension was spotted on multiple petri dishes each containing 20 ml VY2-agar. At desired time points, the agar of 10 plates was harvested and lyophilized. Subsequently, secondary metabolites were extracted from lyophilized agar by methanol extraction. Ambruticin VS-3 in the samples was quantified by HPLC-MS analysis like described previously[49]. To enable an absolute

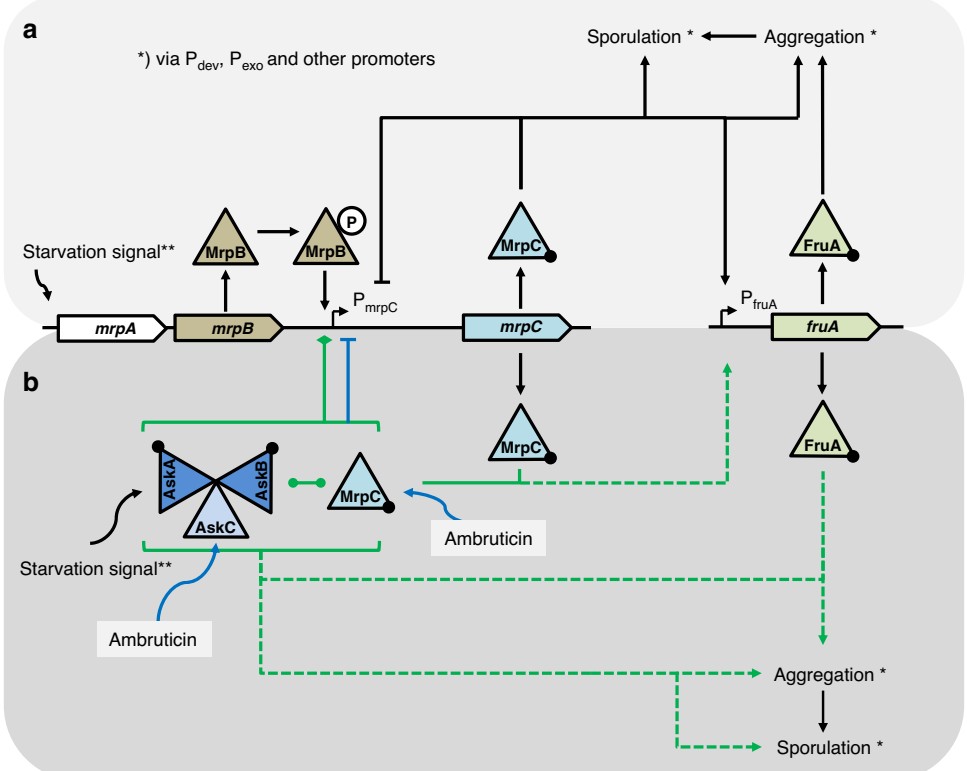

**Fig. 6 Scheme of a revised mrp-module of the *M. xanthus* developmental network.** Pointed boxes: genes. Triangles: proteins. Black circles: DNA-binding ability of a protein. Circled P: phosphorylated protein. **a** Known regulatory cascade of the mrp-module in development. *mrpB* is expressed early in development. After phosphorylation of MrpB, MrpB-P induces expression of *mrpC*, whose expression is then repressed by MrpC via competing with MrpB-P for DNA-binding. Later, MrpC mediated repression of P$_{mrpC}$ is overridden, and MrpC regulates genes necessary for aggregation and sporulation including *fruA*. MrpC and FruA cooperatively regulate the expression of developmental genes. Black arrows: positive regulation. Black lines with bar ends: negative regulation (as summarized by Kroos in 2017[4]). **b** Proposed regulatory cascade of the revised mrp-module in development. At the onset of development, MrpC is tightly bound to P$_{mrpC}$ thereby repressing P$_{mrpC}$ activity. Whether kinase complexes are located at P$_{mrpC}$ at this time point via the observed DNA-binding ability of AskA and/or AskB remains elusive. Eventually, MrpB-P and other starvation-induced signals (\*\*) like cNMP or cyclic-diGMP accumulate and finally lead to a de-repression of P$_{mrpC}$. At this, AskABC plays an important role: The interplay of AskABC or one of its components together with MrpC at P$_{mrpC}$ seems to be the key for successful transition into FB development. At least AskA or AskB is required to enable development via such early de-repression of P$_{mrpC}$. After 18 h of starvation, at least AskB is required to enable further development. At this stage, AskA alone is not sufficient to support ongoing de-repression of P$_{mrpC}$. The most likely effects of AskABC or one of its components on MrpC and its P$_{mrpC}$-repression would be a PTM of MrpC as an effect of Ask kinase activity or a displacement of MrpC at P$_{mrpC}$ by occupation of essential MrpC DNA-binding sites. In addition, a sequestration of MrpC by AskABC cannot be excluded. Subsequently, either PTM-modified MrpC or even the interplay of AskABC-MrpC might induce *fruA* expression. Consequently, FruA and MrpC regulate developmental genes including promoters P$_{dev}$ and P$_{exo}$. At this stage, AskABC or some of its components seem to be of importance as indicated by the need of at least AskB for further development and the observed interaction of FruA and AskABC-components. Green line with circled ends: interplay of MrpC and the kinases. Green line with diamond end: repression or de-repression of promoter P$_{mrpC}$. Dashed green arrows: positive regulation of aggregation and sporulation by released MrpC together with FruA and with a proposed involvement of the kinases. Blue arrows: entry points of ambruticin VS-3 in the GRN. The resulting repression of P$_{mrpC}$ is indicated by a blue line with bar end.

quantification, pure ambruticin VS-3 of known concentration was measured by HPLC-MS in various dilutions. After collection and fragmentation of pseudomolecular ions [M + H] +, the intensities of the fragment ions were summed up and peak integration was carried out. The resulting standard curve was used to determine the overall amount of ambruticin VS-3 per liter of agar.

**Construction of *in-frame* deletion mutants.** All in-frame deletion mutants were obtained as previously reported[50]. Briefly, the sequences upstream and downstream of the regions to be deleted were amplified by overlap extension PCR[51] using Phusion polymerase (ThermoFisher) and cloned into the vector pSWU41[52]. After electroporation, kanamycin resistant (Km$^R$) colonies were selected and grown on CTT without any selective pressure, favoring a second homologous recombination. Cells were subsequently diluted and plated on CTT agar with 5% sucrose. Kanamycin-sensitive and sucrose-resistant colonies were analyzed by PCR for the presence of mutant allele and the absence of the wild type allele.

**Determination of promoter activities using promoter-*lacZ* fusions.** Plasmids derived from pCKβgal_att_ter were constructed and used for quantification of promoter activities via determination of β-galactosidase activity as previously

reported[49]. In summary, cells carrying the desired promoter-*lacZ*-fusion as a result of genomic integration at the attB-site were grown and concentrated to an OD$_{600}$ of 7.5 in CF medium. Subsequently, cells were spotted on CF agar containing different concentrations of ambruticin VS3 or glycerol (see also "growth conditions"). Cells were harvested at different time points during incubation at 30 °C. Cells were resuspended in 200 µl BugBuster protein extraction reagent (Novagen), followed by an incubation of at least 20 min at room temperature to obtain cell lysis. β-galactosidase specific activity was determined by adding 2 µl of the lysates to 98 µl of pure water and 100 µl of 2 x ßgal-buffer (50 mM Tris–HCl pH 7.5, 250 mM NaCl, 4 mM MgCl$_2$, 2 mM 4-methylumbelliferyl-β-D-glucuronide, 1.6 µl/ml ß-mercaptoethanol). Release of 4-methylumbelliferone was recorded at 30 °C using a SpectraMax M5 microplate reader (Molecular Devices; excitation at 365 nm, emission filter 460 nm). Protein concentration of each sample was determined by Bradford-assays[53]. β-galactosidase specific activity was expressed as picomoles of 4-methylumbelliferone released per minute per milligram of protein and plotted against time of sampling during incubation on CF-agar.

**Protein purification.** MrpC, FruA, AskA, AskB and MXAN_0727 were cloned, overexpressed and purified as N-terminally, SUMO-His6-tagged fusion proteins using pE-SUMO3-kan. Coding regions of the proteins were amplified by PCR.

Using respective oligonucleotides (Table S3), restriction site sequences were attached at both ends of the PCR fragments to enable cloning into pE-SUMO3-kan precut using the same restriction enzymes (*Bam*HI and *Kpn*I for MXAN_0727; *Bam*HI and *Xho*I for MrpC and FruA). In addition, AskA, AskB and AskC were cloned, overexpressed and purified as a C-terminally, His6-tagged fusion proteins using pET28C(+). Coding regions of AskB and AskC were amplified by PCR thereby attaching restriction sites for *Pci*I and *Not*I, coding region of AskA was amplified thereby attaching restriction sites for *Bsp*HI and *Xho*I using the respective oligonucleotides. After restriction, *askB* and *askC* amplificates were cloned into pET28C(+) precut with *Nco*I and *Not*I, the *askA* amplificate was cloned into pET28C(+) precut with *Nco*I and *Xho*I. The expression and purification of the recombinant SUMO-His6-tagged- or C-terminally His6-tagged proteins was performed like described earlier including removal of the SUMO-tags of MrpC, FruA and MXAN_0727 except that buffer exchange of proteins was performed using PD10 columns (GE Healthcare)[49]. For respective buffer composition in the assays, refer to MST- and SPR analyses.

**Bacterial Two Hybrid (BACTH) analysis**. Protein interaction studies as preliminary analysis of interaction between proteins were performed using the BACTH system (Euromedex)[54]. The adenylate cyclase domains T25 and T18 were attached to the C-terminus or N-terminus of the proteins of interest by cloning the genes into plasmids provided by the manufacturer (pKT25, pKNT25, pUT18, and pUT18C). In case of each examined domain, both variants, N-terminal and C-terminal fusions were constructed to prevent any false negative result, which might be due to the orientation of the fusion of the domain of interest to T25 or T18. The different combinations to be tested were transformed into *E. coli* BHT101 by electroporation and the resulting colonies were grown and tested for blue color development upon inoculation on M63 plates as described above. Colonies exhibiting a blue color phenotype indicate an interaction of the domains fused to the T25 or T18 reporter domains in vivo.

**Microscale Thermophoresis (MST analysis)**. In order to elucidate both, protein-protein interactions and interactions of proteins and small molecule ligands, a MST analysis was performed[55]. All proteins used in MST were re-buffered into buffer KB1 (500 mM NaCl, 25 mM HEPES, 0.2% Triton X100 v/v, pH 7.4) using PD10 columns (GE Healthcare). C-terminally His6-tagged AskA, AskB and AskC were labeled according to the vendor's manual using the NT-647-His labeling dye of the Monolith NT His-Tag Labeling Kit RED-tris-NTA (NanoTemper) to obtain NT647-AskA, NT647-AskB and NT647-AskC. AskA, AskB, AskC, MrpC, FruA, MXAN_0727, bovine serum albumin (BSA), ambruticin VS3 and myxovirescin A[56] were used as non-labeled ligands. NT647-labeled proteins were titrated with serial 1:1 dilutions of the non-labeled proteins or small molecules in KB1. The interaction of proteins was tested on a NanoTemper Monolith™ NT.115 instrument using standard capillaries at 25 °C, 40% LED power and 40% MST power. The experiments were performed in triplicates. The response values were averaged and plotted against the non-labeled ligand concentration. Using the MO.Affinity Analysis software (NanoTemper), $K_D$ values could be extracted after fitting to the quadratic equation shown in Fig. S2.

**Surface Plasmon Resonance (SPR analysis)**. Protein-protein interactions and interactions of proteins and small molecules were verified in parallel performing an equilibrium analysis using SPR. All proteins used in SPR were re-buffered into buffer KB1 or immobilization buffer (10 mM sodium acetate, pH 4.5). Interaction studies were carried out in triplicates using a Biacore X100 instrument (GE Healthcare) at 25 °C in KB1 as running buffer. Proteins in immobilization buffer, which served as immobilized ligands, were immobilized on a CM5 chip surface (GE Healthcare) according to the amine coupling procedure described by the vendor (GE Healthcare). Molecules in buffer KB1 used as analyte in the experiments were perfused in various concentrations ranging from 0.001 μM to 100000 μM and various contact times from 120 s to 180 s (depending on the pair of molecules under investigation) over both, the reference channel and the detection channel, followed by buffer KB1. Regeneration of the chip surface was achieved by flooding the chip surface using 10 mM glycine pH 1.5 for 600 seconds. Data analysis was performed using the Biacore X100 Evaluation software Version 2.0.1 (GE Healthcare). Briefly, the Rmax values of the obtained sensorgrams were plotted in dependency of the analyte concentration. $K_D$ values were obtained after fitting to the curves assuming a 1:1 interaction. $K_D$ values of the triplicates were averaged and were used together with $K_D$ values from MST analysis to calculate overall $K_D$ values for each interaction (Table 1).

**DNA-protein pull-down assay**. The DNA-protein pull-down assay was performed as described previously[49]. The same promoter region used for analyzing *mrpC* promoter activity was amplified from DK1622 genomic DNA using a pair of 5'-biotinylated primers (Table S3). 1 mg of Dynabeads® M-280 Streptavidin (Invitrogen) was used to bind 12 pmol of the promoter DNA. Cells from *M. xanthus* DK1622 incubated on CF agar with and without 60 ng/ml ambruticin VS3 for 48 h were subjected to sonic disruption. The procedure was subsequently performed like described before[49]. Peptide mass fingerprint

MALDI-MS analysis of single bands revealed the identity of the isolated DNA-binding proteins.

DNA-protein pull-down assays to examine binding of purified proteins AskA, AskB, AskC and MrpC to the promoter region of *mrpC* were performed as follows. The single attempts contained various compositions of AskA, AskB, AskC and MrpC (each protein at 50 μM final) in buffer DP-KNa65 (25 mM HEPES, 60 mM KCl, 65 mM NaCl, 8 mM $MgCl_2$, 10 μM ATP, 25 μg/ml poly-dIdC, 10% glycerol, 0.1% Triton X100, pH 7.4). If desired, ambruticin VS-3 was added to a final concentration of 10 nM. The mixtures were incubated for 10 min, 4 °C on a roller mixer at 40 rpm. 12 pmol of the promoter DNA BTN-$P_{mrpC}$ bound to 0.2 mg Streptavidin Magnetic Beads (NEB) were added to the mixtures. As negative controls, Streptavidin Magnetic beads without any DNA were added in the same amount. After an additional incubation for 5 min as described above, the beads were washed twice with 1000 μl DP-KNa65 supplemented with or without ambruticin VS-3 (4 °C). Additional washing steps were performed using 500 μl, 250 μl and 100 μl DP-KNa65. Elution of proteins bound to the promoter DNA was achieved by adding 25 μl DP-KNa500 buffer (25 mM HEPES, 60 mM KCl, 500 mM NaCl, 8 mM $MgCl_2$, 10 μM ATP, 10% glycerol, 0.1% Triton X100, pH 7.4). The eluted proteins were finally analyzed by SDS-PAGE.

**Protein-protein pull-down assay**. Protein-protein pull-down assays to study the interaction of purified proteins AskA, AskB, AskC and MrpC were performed as follows. The single attempts contained various compositions of AskA, AskB, and MrpC (each protein at 50 μM final) in buffer KB1r (125 mM NaCl, 25 mM HEPES, 0.2% Triton X100 v/v, pH 7.4). If desired, ambruticin VS-3 was added to a final concentration of 10 nM. The mixtures were incubated for 10 min, 4 °C on a roller mixer at 40 rpm. Two micromoles of C-terminally His-tagged AskC attached to 2 mg Dynabeads® His-Tag Isolation & Pulldown magnetic beads were added to the mixtures. As negative controls, beads without C-terminally His-tagged AskC were added in the same amount. After an additional incubation for 5 min as described above, the beads were washed twice with 1000 μl KB1r supplemented with or without ambruticin VS-3 (4 °C). Additional washing steps were performed using 500 μl, 250 μl and 100 μl KB1r. Elution of proteins bound to AskC was achieved by adding 25 μl KB1e (125 mM NaCl, 25 mM HEPES, 500 mM imidazole, 0.2% Triton X100 v/v, pH 7.4). The eluted proteins were finally analyzed by SDS-PAGE.

**Electrophoretic mobility shift assays (EMSA)**. EMSA assays were performed like described before with the following modifications[49]. All proteins used for EMSA analysis were re-buffered into buffer KB1 using PD10 columns. DNA fragments comprising promoter regions of interest were amplified by PCR using the respective primers (Table S2). The oligonucleotides were conjugated either with 5'-hexachloro-fluorescein phosphoramidite (HEX) or with Cy5 at the 5' ends. Two hundred femtomoles of HEX-labeled or Cy5-labeled DNA comprising promoter regions of interest were used in each attempt. In case of a competition assay to decipher a preference of a DNA-binding protein for one specific DNA sequence, both differently labeled DNA fragments were added. Each attempt further contained glycerol (8% v/v), sample buffer (25 mM potassium acetate, 8 mM $MgSO_4$, 0.1 mM DTT, 15 mM Tris–HCl pH 7.3), poly-dIdC (250 ng), proteins of interest (MrpC: 10 picomoles; AskA, AskB and AskC: 3 picomoles) and ambruticin VS-3 (5 nM to 1 μM) in a final volume of 15 μl. Samples were incubated at 30 °C for 30 min. Native gels (0.5x Tris-borate-EDTA, 5% polyacrylamide) were pre-run for 30 min at 20 mA and 4 °C. Subsequently, samples were loaded on the gels and separation of putative complexes was performed at 20 mA, 4 °C for 45–90 min. The fluorescently labeled DNA was visualized using a Typhoon 9410 gel imager (GE Healthcare) using the following settings: HEX, excitation laser at 532 nm, emission filter 555 nm; Cy5, excitation laser at 633 nm, emission filter 670 nm.

**Reporting summary**. Further information on research design is available in the Nature Research Reporting Summary linked to this article.

## Data availability
The authors declare that the data supporting the findings of this study are available within the paper and its supplementary information file. Source data are provided with this paper.

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

## Author contributions

R.M. and C.V. conceived this study. F.M.T. and C.V. constructed and characterized the kinase-deficient strains of *M. xanthus*. F.M.T. performed the BACTH assays. F.M.T. and C.V. performed SPR and MST analyses. C.V. performed EMSA assays. C.V. performed DNA-protein and protein-protein pull-down experiments. C.V. performed coincubations assays of myxobacteria. C.V. and F.M.T. wrote the manuscript.

## Funding

## Competing interests

The authors declare no competing interests.
