## [Peer Review File · Nature Communications]

Reviewers' comments:

Reviewer #1 (Remarks to the Author):

The authors report that the antifungal compound ambruticin produced by the myxobacterium *Sorangium cellulosum* inhibits starvation-induced multicellular development of another myxobacterium, *Myxococcus xanthus*, providing an example of interspecies communication. Under these conditions, ambruticin prevents formation of multicellular fruiting bodies and causes partial differentiation of cells into "pseudospores" that resist sonication but germinate poorly. Interestingly, glycerol at a much higher concentration has a similar effect. Because glycerol has been known to induce sporulation in the presence of nutrients, the authors tested ambruticin and found that a much lower concentration induces sporulation. These chemically-induced spores resist sonication and mild heating, and germinate efficiently. The low concentration of ambruticin required and the fact that it is produced by another myxobacterium in the soil niche suggests that ambruticin is a natural, biologically-relevant inducer of *M. xanthus* sporulation, unlike glycerol, solving a long-standing puzzle. Moreover, the inhibition of starvation-induced fruiting body sporulation and the induction of sporulation in the presence of nutrients may allow ambruticin-producing *S. cellulosum* to outcompete *M. xanthus* under both starvation and growth conditions in the soil environment, a point the authors may wish to make.

The authors also report some molecular insights into the response of *M. xanthus* to ambruticin. Analysis of the *S. cellulosum* ambruticin biosynthetic gene cluster revealed a hybrid histidine kinase similar to *Candida albicans* NIK1, which has been implicated in susceptibility to ambruticin. These two kinases are similar to two *M. xanthus* kinases, which the authors name AskA and AskB. Deletion of askA or askB had no effect, but deletion of both inhibited starvation-induced fruiting body formation and caused pseudospores to form in the absence of ambruticin, much like the effect of ambruticin on wild type. This shows that AskA or AskB must be present to allow fruiting body development and prevent pseudospore formation. The results suggested that ambruticin might inhibit both AskA and AskB (which the authors may wish to state explicitly to help readers follow the rationale), but ambruticin did not appear to bind directly to either kinase. A third hybrid histidine kinase, AskC, is located immediately downstream of AskB. Deletion of askC rendered *M. xanthus* insensitive to ambruticin, which bound directly to AskC, but the binding was weak compared with the effective ambruticin concentration on wild-type cells. Nevertheless, the results implicate the three Ask kinases in the response to ambruticin.

The authors go on to show pairwise interactions among the three Ask kinases, then they show that ambruticin and the Ask kinases affect mrpC transcription. MrpC is a transcription factor required for fruiting body development, including for transcription of a second key developmental transcription factor called FruA. Interestingly, both MrpC and FruA interact pairwise with each Ask kinase in vitro. Also, ambruticin, at a low concentration comparable to that affecting fruiting body formation, enhances MrpC binding to mrpC promoter region DNA in vitro, which based on other work likely represses mrpC transcription. Finally, AskB appears to bind specifically to mrpC promoter region DNA in vitro.

The authors attempt to integrate their results into models describing how the Ask kinases affect fruiting body development and how ambruticin affects mrpC transcription.

Overall, the authors have some very interesting observations. The identification of ambruticin and the characterization of its phenotypic effects on wild-type *M. xanthus* development, as well as on ask mutants, are the strongest results. The broad significance and novelty of the interspecies competition uncovered by the work needs to be better-articulated – *S. cellulosum* via ambruticin appears to manipulate the developmental programs of *M. xanthus* in order to outcompete under different conditions. The insights into the molecular mechanisms are not as strong. The genetic analysis of ask mutants could go farther, as suggested below. Evidence for more than two components interacting is lacking, and none of it is in *M. xanthus*. By ignoring this, the authors

overstate their results repeatedly throughout the manuscript by referring to protein complexes containing the three Ask kinases, MrpC, and FruA. Hence, the model in Fig. 5b describing how the Ask kinases affect fruiting body development is highly speculative and not very informative. The model in Fig. 5c describing how ambruticin affects *mrpC* transcription is reasonable given the *in vitro* results, but lacks supporting *in vivo* results (e.g., ChIP analysis showing the Ask kinases bound to the *mrpC* promoter region). Also, if ambruticin binding to AskC is the first step as proposed, the relatively weaker binding of ambruticin *in vitro* to AskC as compared with MrpC, does not explain the very low concentration of ambruticin that can inhibit fruiting body formation.

Major comments:

1. While the effects of ambruticin on *M. xanthus* documented in Fig. 1 are impressive, what is the evidence that ambruticin is responsible for the effect of *S. cellulosum* on *M. xanthus* shown in Fig. 1a? Many questions related to interspecies communication are not addressed: How much ambruticin is made by *S. cellulosum*? How is it released? Importantly, what extracellular concentration is achieved? While ambruticin is small enough to pass through outer membrane protein pores, it has carboxyl and hydroxyl groups on one end, so how does it get through the *M. xanthus* inner membrane? The authors may not be able to address all these questions, but a measure of the extracellular concentration seems crucial.

2. The genetic analysis of ask mutants is incomplete. The analysis of the *askA askB* double mutant in Fig. 2ab and the *askC* single mutant in Fig. 3ab, together with binding of ambruticin to AskC *in vitro* (Table 1), suggest that ambruticin-bound AskC inhibits both AskA and AskB, blocking fruiting body formation and causing pseudospores to form. This model is not stated in the manuscript, but the model makes predictions about the phenotypes of *askA askC*, *askB askC*, and *askA askB askC* mutants in the presence or absence of ambruticin, which should be tested. The results would likely provide some additional insight into the molecular mechanisms of Ask kinase function during starvation-induced development.

3. Related to the preceding point, the analysis of ask mutants with respect to chemically-induced sporulation is even less complete. The *askA* and *askB* single mutants, as well as the additional double and triple mutants mentioned above, should be tested with both high ambruticin and high glycerol. Interestingly, neither treatment induced sporulation of the *askA askB* double mutant (Fig. 2d). In contrast, neither AskA nor AskB is necessary for ambruticin or glycerol (at lower concentrations) to block starvation-induced fruiting body formation and cause pseudospores to form (Fig. 2ab). Hence, AskA and/or AskB function differently under the two developmental conditions. This may confuse some readers and it is not well-explained in the manuscript. The authors go on to show that the *askA askB* double mutant fails to express *PmrpC-lacZ* during starvation-induced development (Fig. S6). Given that *mrpC* has been shown to be up-regulated and required during glycerol-induced sporulation, the authors should test whether *mrpC* is required for ambruticin-induced sporulation.

4. The authors provide evidence for pairwise interactions among AskA, AskB, and AskC using two *in vitro* approaches, and for interaction between the receiver domains of AskA and AskC using two-hybrid analysis in *E. coli*. Lacking is evidence that the three proteins form a complex *in vitro* (i.e., interact simultaneously rather than forming mutually exclusive pairs) or in *M. xanthus* (e.g., using a pull-down approach). Obtaining such evidence would strengthen the molecular analysis and is necessary if the authors want to refer to protein complexes containing the three Ask kinases, MrpC, and FruA, other than in hypothetical terms.

5. The authors used *lacZ* fusions to four genes to examine the effects of ask mutations, ambruticin, and glycerol on starvation-induced developmental gene expression (Fig. S6). Expression from three promoters (*mrpC*, *fruA*, *dev*) was blocked in the *askA askB* double mutant and upon addition of ambruticin or glycerol to wild type. The effects of ambruticin and glycerol were relieved in the *askC* single mutant, consistent with the model mentioned above that

ambruticin binds to AskC and this complex inhibits both AskA and AskB, blocking fruiting body formation and causing pseudospores to form. The implication is that very little or no expression of *mrpC* and *fruA* is required for pseudospore formation, suggesting it differs from starvation- or chemically-induced sporulation. These points are worth mentioning. Also noteworthy is that neither ambruticin nor glycerol prevented a high level of *mrpC-lacZ* expression in the *mrpC* mutant. This is important because it suggests that the early events of the starvation-induced developmental pathway, including production of phosphorylated MrpB which activates *mrpC* transcription, occur in the presence of ambruticin or glycerol. Moreover, the data shows that MrpC and AskC are necessary for ambruticin or glycerol to inhibit *mrpC-lacZ* expression. The authors go on to show that ambruticin, at a low concentration comparable to that affecting fruiting body formation, enhances MrpC binding to *mrpC* promoter region DNA in vitro (Fig. 4b). Hence, a simple model would be that ambruticin makes MrpC a better repressor of *mrpC* transcription, but this would not explain the requirement for AskC to inhibit *mrpC* transcription. The authors propose a model in Fig. 5c in which ambruticin binding to AskC is the first step, but as mentioned above the relatively weaker binding of ambruticin in vitro to AskC as compared with MrpC is inconsistent with this model and the model does not explain the very low concentration of ambruticin that can inhibit fruiting body formation.

6. The binding of the AskB kinase to the *mrpC* promoter region (Fig. 4c) is a novel observation of potential broad significance, so testing in vivo (e.g., using ChIP) is needed.

Additional comments:

The Abstract overstates the evidence (l. 21-25). As noted above, there is no evidence for the AskABC complex, let alone larger complexes that include FruA and/or MrpC, although there is evidence of pairwise interactions among all these components. Also, testing the binding of the Ask kinases to the *mrpC* promoter region in vivo is needed to support "a DNA-binding complex of kinases".

One or more references regarding (p)ppGpp involvement should be cited (l. 39).

What is the evidence that *M. xanthus* evolved a mechanism to sporulate under nutrient-rich conditions (l. 43)? Do they encounter 0.5 M glycerol? They do presumably encounter ambruticin, it seems preferable to mention evolution of a mechanism to sporulate under nutrient-rich conditions later in the manuscript.

What does "they" refer to (l. 48)? "effects" or "inducers"? Genes like *exo* and *nfs* seem to be involved in both responses.

(l. 57-59) overstates the evidence, as explained for the Abstract above. Elsewhere, the authors need to be more careful referring to protein complexes they have not demonstrated.

Does 4 nM ambruticin induce pseudospores? Why was 20 nM used in Fig. 1d, 60 nM in Fig. 1e, and 100 nM in Fig. 1f? Do the spores in Fig. 1f germinate efficiently? Line 87 mentions the "rate" but the data is not shown.

Should "Both" be "Either" on line 115?

The scale bars seem too long to represent 1 μ m in Fig. 2b and 2d.

(l. 126 and elsewhere) using AskAB or AskABC as abbreviations for the separate proteins is unacceptable. For example, this also leads to confusion on l. 143 and as noted above the experiments showing pairwise interactions do not prove the existence of a three-membered complex.

Fig. S2b – use of subscripts is not uniform in the equation.

(l. 151, Table 1, and elsewhere including supplemental figures) – too many significant figures are reported considering the large standard deviations.

exo is referred to as an “endpoint” operon (l. 160) and component (l. 162). What does this mean? It does not behave as if it acts downstream in the regulatory network (Fig. S6). The y-axis for Pexo-lacZ in Fig. S6b should be expanded since the lines cannot be distinguished.

Up regulation of PmrpC and Pexo in the mrpC mutant suggests an auto-inhibitory effect on PmrpC, but not on Pexo (up regulation of Pexo in an exo mutant would imply auto inhibition).

Fig. 4a – what is the protein at about 55 kDa? What does PAA mean in the legend? Is it not SDS-PAGE?

(l. 185) “Obviously, alteration of MrpC binding affinity to PmrpC is the main causal effect of ambruticin inhibition of development.” overstates what can be concluded from Fig. 4a.

(l. 209-213) makes statements that are cryptic, unsupported, and not informative, so this should be deleted.

(l. 244-246) is unclear. The data in Fig. S6 show that AskC is involved in starvation-induced developmental gene expression in the presence of ambruticin. Why do the authors say “but only AskAB is involved in development”? What does “additional entry points into the GRN that might be mirrored” mean? Presumably, the GRN for chemically-induced sporulation is different from that for starvation-induced multicellular development. Related to this, what does “AskAB are essential for development whereas AskABC is essential for sporulation what is highlighted by different colors.” mean in the Fig. 5b legend (l. 701)?

(l. 260-262) and the model in Fig. 5c ignore the point mentioned above that ambruticin binds weakly to AskC in vitro. Are the authors proposing that ambruticin binds with higher affinity to AskABC? If so, this should be tested.

Should “addressed” be “affected” on l. 300?

Table S2 – reference 3 is missing and subsequent references are misnumbered.

Something is missing between “perfusing” and “data” on l. 413.

Reviewer #2 (Remarks to the Author):

The manuscript by Marcos-Torres et al describes how one myxobacteria species inhibits another’s ability to undergo fruiting body development by a diffusible factor. They identify the causative agent, a secondary metabolite called ambruticin, which was previously shown to have antifungal activity. The potency of ambruticin is striking, 4 nM, suggesting this activity is physiologically relevant. Interestingly they also show that ambruticin induces sporulation by the classic glycerol induction pathway during non-starvation conditions. They then discovered three histidine kinases as playing a role in the ambruticin/glycerol pathways and that a key developmental regulator, MrpC, is regulated by ambruticin. Overall this study reports interesting findings and much of the experimental work is carefully done and impressive. Their findings also add depth and complexities to central aspects of the ‘gene regulatory network’ (GRN) controlling development in Myxococcus

xanthus. However, there are missing pieces and how their findings fit into a cogent model still needs to be explored and resolved. Additionally, the text needs to be polished; there are numerous grammatical errors and the logic flow at times is difficult to follow. Some examples are given below.

General points:

Does ambruticin block development of other myxobacteria isolates/species? If so, this would strengthen the authors' argument that this activity is indeed physiologically/ecologically relevant.

To aid the reader, the chemical structure of ambruticin should be provided, preferably in Fig. 1.

Table 1 and elsewhere, how is ambVS3 (ambruticin VS3) different than ambruticin and why are the results for MrpC binding different?

Lines 98-113 and Fig. S1 discusses/shows the domain architecture of various HK with HAMP domains, but the text lacks a clear explanation of what HAMP domains do and how they might function in these particular proteins. From the literature, HAMP motifs are typically found immediately downstream of transmembrane helices and serve as communication modules between the input and output domains of HK. This is not discussed or whether these particular HK have TM helices. Additionally, it is very unusual for proteins to have many (e.g. 16) HAMP motifs, which should also be discussed.

In Fig. S2, explain why some of the titration curves go down while others go up.

Fig. 4a. Legend states all bands were identified by MALD-Tof, therefore please state the identity of the other protein band running at 55 kDa.

Fig. 4a,b. The impact of Amb addition on MrpC is rather potent and intriguing. What happens if glycerol is instead added to cells in these experiments?

Fig. 4c. The finding that AskB binds specifically to the mrpC promoter is interesting. By extension from the results shown in

Fig. 4ab, what happens to the AskB DNA binding activity when ambruticin is added to cells prior to extract preparation? Also, when looking at the Fig. 1S cartoon structure of AskB it is not clear where a DNA binding domain could be and, moreover, as discussed, this is a unique finding that a HK binds DNA. It would therefore be informative to do structure-function studies to identify the DNA binding domain within AskB. Finally, the competitor DNA (Pmxan_0552) was used at equal amounts; what happens when competitor DNA is used in large excess, e.g. 100-fold?

Line 193: Since the three kinases (AskABC) bind to MrpC and FruA, please discuss what is known about these latter proteins being phosphorylated.

Since ambruticin is an antifungal agent the question arises whether it has antibacterial activity on *M. xanthus*. In this regard, does ambruticin block the growth of an askC mutant (which is refractive to sporulation induction)?

Prior work by O'Connor & Zusman showed that induction of the glycerol spore pathway involves chromosomal induction of beta-lactamase in response to agents that damage the peptidoglycan, as well as, occurring naturally during fruiting body development. The developmental transition from vegetative cells to spores obviously involves major morphological changes and re-organization of the peptidoglycan, and agents that damage the peptidoglycan apparently induce sporulation independently of starvation. This work is relevant and accordingly should be discussed.

Some of the experiments and interpretations need better explanations. For example, the cartoons

in Fig. S5 seem to indicate protein fusions were created where the REC or REC3 domains were placed up or downstream of the p18/25 domains. If this is correct it should be described in the legend and/or Result section and why it was done.

Minor points:

The last sentence of the abstract is confusing and needs to be re-worded.

Line 40: "AEC-signals" is a confusing abbreviation because it involves three signals, but reads as one. Change to "A, E and C-signals" or in a similar way to make the distinction.

Line 42: change "influencing" to "controlling"

Line 55: change "effective" to "potent"

Line 56: add commas before and after "AskABC"

Line 586: after "48" add "h"

The second paragraph, lines 114-130, is terse and difficult to follow. More rationale and data interpretation is needed, e.g. provide a preface to AskA and AskB binding to each other.

Line 133, after "glycerol" add "in rich medium" or "under starvation conditions" or re-write sentence, so the results can be clearly understood.

Line 175: insert "reporter" after "all" to qualify statement.

Line 186: The statement "is the main causal effect" is too strong as other cellular factors could also play a role. Change to something like "could provide an explanation for why ambruticin...."

Fig. S4 legend, line 113-114, please clarify sentence. What does "...any other protein in the absence of DNA..." mean?

Line 207-8: change "...we performed initial additional EMSA assays, also including the kinases" to "we performed preliminary EMSA assays that also included the kinases." The next three sentences should be deleted or re-worded because they are vague/not informative.

Dear reviewers,

We want to thank all the reviewers for the intense reading and for the helpful comments related to our first manuscript. We performed numerous additional experiments to meet your remarks and suggestions. All the results are now included in the manuscript. However, due to the additional results we had to rewrite parts of the manuscript and had to make changes regarding the composition of Figures.

Reviewers' comments:

Reviewer #1 (Remarks to the Author):

The authors report that the antifungal compound ambruticin produced by the myxobacterium *Sorangium cellulosum* inhibits starvation-induced multicellular development of another myxobacterium, *Myxococcus xanthus*, providing an example of interspecies communication. Under these conditions, ambruticin prevents formation of multicellular fruiting bodies and causes partial differentiation of cells into “pseudospores” that resist sonication but germinate poorly. Interestingly, glycerol at a much higher concentration has a similar effect. Because glycerol has been known to induce sporulation in the presence of nutrients, the authors tested ambruticin and found that a much lower concentration induces sporulation. These chemically-induced spores resist sonication and mild heating, and germinate efficiently. The low concentration of ambruticin required and the fact that it is produced by another myxobacterium in the soil niche suggests that ambruticin is a natural, biologically-relevant inducer of *M. xanthus* sporulation, unlike glycerol, solving a long-standing puzzle. Moreover, the inhibition of starvation-induced fruiting body sporulation and the induction of sporulation in the presence of nutrients may allow ambruticin-producing *S. cellulosum* to outcompete *M. xanthus* under both starvation and growth conditions in the soil environment, a point the authors may wish to make.

ANSWER of the authors: This point has been added to the text.

The authors also report some molecular insights into the response of *M. xanthus* to ambruticin. Analysis of the *S. cellulosum* ambruticin biosynthetic gene cluster revealed a hybrid histidine kinase similar to *Candida albicans* NIK1, which has been implicated in susceptibility to ambruticin. These two kinases are similar to two *M. xanthus* kinases, which the authors name AskA and AskB. Deletion of askA or askB had no effect, but deletion of both inhibited starvation-induced fruiting body formation and caused pseudospores to form in the absence of ambruticin, much like the effect of ambruticin on wild type. This shows that AskA or AskB must be present to allow fruiting body development and prevent pseudospore formation. The results suggested that ambruticin might inhibit both AskA and AskB (which the authors may wish to state explicitly to help readers follow the rationale), but ambruticin did not appear to bind directly to either kinase.

ANSWER of the authors: As described further below, the contribution of the Ask kinases to the effects has further been investigated. Previously missing combinations of in-frame deletion

mutants were constructed and analyzed (double- as well as triple mutants). The results are now presented and discussed in the manuscript.

A third hybrid histidine kinase, AskC, is located immediately downstream of AskB. Deletion of askC rendered *M. xanthus* insensitive to ambruticin, which bound directly to AskC, but the binding was weak compared with the effective ambruticin concentration on wild-type cells. Nevertheless, the results implicate the three Ask kinases in the response to ambruticin.

The authors go on to show pairwise interactions among the three Ask kinases, then they show that ambruticin and the Ask kinases affect mrpC transcription. MrpC is a transcription factor required for fruiting body development, including for transcription of a second key developmental transcription factor called FruA. Interestingly, both MrpC and FruA interact pairwise with each Ask kinase in vitro. Also, ambruticin, at a low concentration comparable to that affecting fruiting body formation, enhances MrpC binding to mrpC promoter region DNA in vitro, which based on other work likely represses mrpC transcription. Finally, AskB appears to bind specifically to mrpC promoter region DNA in vitro.

The authors attempt to integrate their results into models describing how the Ask kinases affect fruiting body development and how ambruticin affects mrpC transcription.

Overall, the authors have some very interesting observations. The identification of ambruticin and the characterization of its phenotypic effects on wild-type *M. xanthus* development, as well as on ask mutants, are the strongest results. The broad significance and novelty of the interspecies competition uncovered by the work needs to be better-articulated – *S. cellulorum* via ambruticin appears to manipulate the developmental programs of *M. xanthus* in order to outcompete under different conditions. The insights into the molecular mechanisms are not as strong. The genetic analysis of ask mutants could go farther, as suggested below. Evidence for more than two components interacting is lacking, and none of it is in *M. xanthus*. By ignoring this, the authors overstate their results repeatedly throughout the manuscript by referring to protein complexes containing the three Ask kinases, MrpC, and FruA. Hence, the model in Fig. 5b describing how the Ask kinases affect fruiting body development is highly speculative and not very informative. The model in Fig. 5c describing how ambruticin affects mrpC transcription is reasonable given the in vitro results, but lacks supporting in vivo results (e.g., ChIP analysis showing the Ask kinases bound to the mrpC promoter region). Also, if ambruticin binding to AskC is the first step as proposed, the relatively weaker binding of ambruticin in vitro to AskC as compared with MrpC, does not explain the very low concentration of ambruticin that can inhibit fruiting body formation.

Major comments:

1. While the effects of ambruticin on *M. xanthus* documented in Fig. 1 are impressive, what is the evidence that ambruticin is responsible for the effect of *S. cellulorum* on *M. xanthus* shown in Fig. 1a? Many questions related to interspecies communication are not addressed: How much ambruticin is made by *S. cellulorum*? How is it released? Importantly, what extracellular concentration is achieved? While ambruticin is small enough to pass through outer membrane

protein pores, it has carboxyl and hydroxyl groups on one end, so how does it get through the *M. xanthus* inner membrane? The authors may not be able to address all these questions, but a measure of the extracellular concentration seems crucial.

ANSWER of the authors: The extracellular concentration of ambruticin during a 6 days incubation period of *S. cellulosum* has been determined and is now included in the text/figures. The concentration is at least 80-fold higher than the 4nM used in the assays employing purified compound (see now in Fig.1). Up to date, nothing is known how this substance is released by the producer or how it is subsequently taken up by the target cell (*M. xanthus*). This is subject of future work and will be addressed. In both cases, it is likely that this transport is not achieved by diffusion.

All observations account for ambruticin to be the causative agent. Only ambruticin-producers cause this phenotype in *M. xanthus* (see now also photographs of coincubations of *M. xanthus* with a non-ambruticin producing myxobacterial strain in Fig. 1), the extracellular concentration of ambruticin exceeds the concentration of tested pure compound, and the observations made by addition of pure ambruticin equal the observations made during coincubations with the ambruticin producing *S. cellulosum*. The authors do not see any possible scenario of ambruticin not being the causative compound.

2. The genetic analysis of ask mutants is incomplete. The analysis of the askA askB double mutant in Fig. 2ab and the askC single mutant in Fig. 3ab, together with binding of ambruticin to AskC in vitro (Table 1), suggest that ambruticin-bound AskC inhibits both AskA and AskB, blocking fruiting body formation and causing pseudospores to form. This model is not stated in the manuscript, but the model makes predictions about the phenotypes of askA askC, askB askC, and askA askB askC mutants in the presence or absence of ambruticin, which should be tested. The results would likely provide some additional insight into the molecular mechanisms of Ask kinase function during starvation-induced development.

ANSWER of the authors: Indeed, this analysis was incomplete. The lacking mutants have been constructed (Δ askAC, Δ askBC, Δ askABC) and have been characterized. The results are now included. At this, Figures 3 and S6 give an overview of the phenotype of Δ askAC, Δ askBC, Δ askABC and the promoter activity PmrpC in strains Δ askAC, Δ askBC, Δ askABC, Δ askA, Δ askB.

3. Related to the preceding point, the analysis of ask mutants with respect to chemically-induced sporulation is even less complete. The askA and askB single mutants, as well as the additional double and triple mutants mentioned above, should be tested with both high ambruticin and high glycerol. Interestingly, neither treatment induced sporulation of the askA askB double mutant (Fig. 2d). In contrast, neither AskA nor AskB is necessary for ambruticin or glycerol (at lower concentrations) to block starvation-induced fruiting body formation and cause pseudospores to form (Fig. 2ab). Hence, AskA and/or AskB function differently under the two developmental conditions. This may confuse some readers and it is not well-explained in the manuscript. The authors go on to show that the askA askB double mutant fails to express PmrpC-lacZ during starvation-induced development (Fig. S6). Given that mrpC has been shown to be up-regulated and required during glycerol-induced sporulation,

the authors should test whether *mrpC* is required for ambruticin-induced sporulation.

ANSWER of the authors: The analysis of all ask mutants with respect to chemically-induced sporulation has been performed (now including all mutants). We can thus present a more detailed picture of which proteins are essential or which interplay of proteins leads to which phenotype.

The *mrpC* mutant has already been tested and did not sporulate, hence MrpC is required for ambruticin-induced sporulation (data not shown in the manuscript but now mentioned).

4. The authors provide evidence for pairwise interactions among AskA, AskB, and AskC using two in vitro approaches, and for interaction between the receiver domains of AskA and AskC using two-hybrid analysis in *E. coli*. Lacking is evidence that the three proteins form a complex in vitro (i.e., interact simultaneously rather than forming mutually exclusive pairs) or in *M. xanthus* (e.g., using a pull-down approach). Obtaining such evidence would strengthen the molecular analysis and is necessary if the authors want to refer to protein complexes containing the three Ask kinases, MrpC, and FruA, other than in hypothetical terms.

ANSWER of the authors: The studies of pairwise interactions have now been completed with a pull-down approach (see Fig.4). Using biotinylated DNA bearing promoter P_{mrpC} (BTN-P_{mrpC}) and attached to magnetic beads, MrpC alone could be isolated. Using attempts where only AskA, AskB or AskC were present in the mixture, it was clearly visible that indeed both, AskA and AskB bind to the promoter of *mrpC*. AskC could not be fished in absence of AskA or AskB. In contrast, in presence of AskA and AskB, AskC could be isolated using BTN-P_{mrpC} indicating the existence of a complex comprising all three kinases. This could be observed in presence and absence of MrpC. These observations were also independent of ambruticin. Binding of MrpC seems not to be affected by the presence of the kinases. These results are qualitative in vitro results. Based on these in vitro data we reason that besides pairwise interactions of the kinases a protein complex comprising all three kinases should also exist *in vivo* at some point. Certainly, this has to be proven in subsequent work. These data are now presented and discussed correspondingly.

5. The authors used lacZ fusions to four genes to examine the effects of ask mutations, ambruticin, and glycerol on starvation-induced developmental gene expression (Fig. S6). Expression from three promoters (*mrpC*, *fruA*, *dev*) was blocked in the *askA askB* double mutant and upon addition of ambruticin or glycerol to wild type. The effects of ambruticin and glycerol were relieved in the *askC* single mutant, consistent with the model mentioned above that ambruticin binds to AskC and this complex inhibits both AskA and AskB, blocking fruiting body formation and causing pseudospores to form. The implication is that very little or no expression of *mrpC* and *fruA* is required for pseudospore formation, suggesting it differs from starvation- or chemically-induced sporulation. These points are worth mentioning. Also noteworthy is that neither ambruticin nor glycerol prevented a high level of *mrpC-lacZ* expression in the *mrpC* mutant. This is important because it suggests that the early events of the starvation-induced developmental pathway, including production of phosphorylated MrpB which activates *mrpC* transcription, occur in the presence of ambruticin or glycerol. Moreover, the data shows that MrpC and AskC are necessary for ambruticin or glycerol to

inhibit *mrpC-lacZ* expression. The authors go on to show that ambruticin, at a low concentration comparable to that affecting fruiting body formation, enhances MrpC binding to *mrpC* promoter region DNA *in vitro* (Fig. 4b). Hence, a simple model would be that ambruticin makes MrpC a better repressor of *mrpC* transcription, but this would not explain the requirement for AskC to inhibit *mrpC* transcription. The authors propose a model in Fig. 5c in which ambruticin binding to AskC is the first step, but as mentioned above the relatively weaker binding of ambruticin *in vitro* to AskC as compared with MrpC is inconsistent with this model and the model does not explain the very low concentration of ambruticin that can inhibit fruiting body formation.

ANSWER of the authors: The model has been revised and simplified accordingly and is now based on the experimental results only.

6. The binding of the AskB kinase to the *mrpC* promoter region (Fig. 4c) is a novel observation of potential broad significance, so testing *in vivo* (e.g., using ChIP) is needed.

ANSWER of the authors: Indeed this is a novel observation of potential broad significance. We feel the same way like reviewer #1. ChIP is a powerful technique, which was not available in the past. However, a vast number of DNA-binding proteins were initially identified using the experimental setup also presented in the manuscript. We have in the meantime performed additional pull-down experiments. Again, AskB exhibited binding to DNA. In addition, AskA was also able to bind to DNA in these experiments (not in EMSA). A re-investigation of the putative domains present in both proteins revealed a strong likelihood that both proteins possess a large number of putative partial HTH_6_RpiR and partial SpoIIID like domains, which comprise HTH_DEOR_1-like domains. These putative domains occur pairwise overlapping with the HAMP domains in their unique N-termini (see also response to next comment; shown in figure S1). Such domains could not be found in other group III HHK like Nik1. AskC is also missing such putative domains and most likely therefore does not bind to DNA. We also did an additional EMSA introducing a 300-fold excess of non-labeled P*mrpC*-DNA and by doing this could abolish the observed interaction to HEX-labeled P*mrpC*-DNA. Therefore, we strongly believe that we present sufficient *in vitro* data now to state that AskB binds to P*mrpC*-DNA *in vitro*. This is on the one hand no longer an overstatement but also takes into account that a technique like ChIP is available now and will be used in the future to finally confirm the observations *in vivo*. However, this work aiming at specifically addressing this new kind of interaction is out of the scope of the current manuscript. Here, the compelling *in vitro* evidence is highlighted whenever necessary.

Additional comments:

The Abstract overstates the evidence (l. 21-25). As noted above, there is no evidence for the AskABC complex, let alone larger complexes that include FruA and/or MrpC, although there is evidence of pairwise interactions among all these components. Also, testing the binding of the Ask kinases to the *mrpC* promoter region *in vivo* is needed to support “a DNA-binding complex of kinases”.

ANSWER of the authors: Indeed, compelling evidence for this complex was lacking in the previous version of the manuscript. We performed pull-down experiments (protein-protein as well as DNA-protein pull-down experiments). At this, we could show that *in vitro* AskABC form a complex and that at least one of these proteins in the complex simultaneously interacts with MrpC. In addition, the DNA binding ability of AskB could again be seen in the pull-downs. In addition, AskA was able to interact with DNA in the pull-down. Together with the numerous observed putative partial, as well as complete HTH_6 and SpoIIID-like domains in the N-termini of AskA and AskB (overlapping with the HAMP domains) we are convinced that stating that at least AskB is able to specifically interact with DNA comprising PmrpC *in vitro* is reasonable. Additional work to confirm this further *in vivo* is clearly required but will be a different story. This has been stated now explicitly throughout the whole manuscript.

One or more references regarding (p)ppGpp involvement should be cited (l. 39).

ANSWER of the authors: The references have been added.

What is the evidence that *M. xanthus* evolved a mechanism to sporulate under nutrient-rich conditions (l. 43)? Do they encounter 0.5 M glycerol? They do presumably encounter ambruticin, it seems preferable to mention evolution of a mechanism to sporulate under nutrient-rich conditions later in the manuscript.

ANSWER of the authors: This has been changed respectively.

What does “they” refer to (l. 48)? “effects” or “inducers”? Genes like *exo* and *nfs* seem to be involved in both responses.

ANSWER of the authors: The wording has been changed to “these effectors”.

(l. 57-59) overstates the evidence, as explained for the Abstract above. Elsewhere, the authors need to be more careful referring to protein complexes they have not demonstrated.

ANSWER of the authors: Based on the additional results presented in the new version of the manuscript, the wording has been adjusted.

Does 4 nM ambruticin induce pseudospores? Why was 20 nM used in Fig. 1d, 60 nM in Fig. 1e, and 100 nM in Fig. 1f? Do the spores in Fig. 1f germinate efficiently? Line 87 mentions the “rate” but the data is not shown.

ANSWER of the authors: Four nM ambruticin VS-3 is sufficient to induce pseudospores, but as can be seen in the fruiting bodies picture (4nM in Fig 1b), the phenotype is not completely expressed (mounds are still visible indicating an initial aggregation of cells). Consequently, we used 20nM ambruticin VS-3 for further qualitative analysis, e.g. for pictures of development, as this is triggering a full response which can be seen in Fig 1b. As the optimal glycerol concentration required for chemically induced sporulation on rich medium is 5 times the amount required for fruiting body inhibition on plate, we used the same ratio for induction with ambruticin (i.e. 100 nM). To avoid experimental errors derived from effector-concentrations too

close to the minimal requirements we used concentrations that were 3 times the minimal requirements (i.e. 60 nM ambruticin and 300 mM glycerol) in the quantitative analysis such as sporulation or during growth of the strains bearing promoter-*lacZ* fusions. Respective comments have been added in the figure legends.

We have added the rate values of germination in Fig. 1h.

Should “Both” be “Either” on line 115?

ANSWER of the authors: “Both” has been changed to “either”.

The scale bars seem too long to represent 1 μ m in Fig. 2b and 2d.

ANSWER of the authors: This has been changed.

(l. 126 and elsewhere) using AskAB or AskABC as abbreviations for the separate proteins is unacceptable. For example, this also leads to confusion on l. 143 and as noted above the experiments showing pairwise interactions do not prove the existence of a three-membered complex.

ANSWER of the authors: These abbreviations have been changed and separate proteins have been mentioned accordingly.

Fig. S2b – use of subscripts is not uniform in the equation.

ANSWER of the authors: The use of subscripts is uniform now.

(l. 151, Table 1, and elsewhere including supplemental figures) – too many significant figures are reported considering the large standard deviations.

ANSWER of the authors: We are sorry but do not understand the critique in this sentence. Data obtained from SPR indeed exhibited much less standard deviations than those obtained from MST. Nevertheless, even the MST data are significant as confirmed by the evaluation software (NanoTemper).

exo is referred to as an “endpoint” operon (l. 160) and component (l. 162). What does this mean? It does not behave as if it acts downstream in the regulatory network (Fig. S6). The y-axis for Pexo-*lacZ* in Fig. S6b should be expanded since the lines cannot be distinguished.

ANSWER of the authors: “endpoint” has been deleted. The y-axis has been expanded.

Up regulation of PmrpC and Pexo in the *mrpC* mutant suggests an auto-inhibitory effect on PmrpC, but not on Pexo (up regulation of Pexo in an *exo* mutant would imply auto inhibition).

ANSWER of the authors: This has been corrected.

Fig. 4a – what is the protein at about 55 kDa? What does PAA mean in the legend? Is it not SDS-

PAGE?

ANSWER of the authors: The prominent band at 55kDa in both attempts was identified to be a peroxiredoxin-like protein what is now mentioned (AhpC; MXAN1564). PAA=polyacrylamide, it has been replaced by SDS-PAGE.

(l. 185) “Obviously, alteration of MrpC binding affinity to PmrpC is the main causal effect of ambruticin inhibition of development.” overstates what can be concluded from Fig. 4a.

ANSWER of the authors: We changed this into “Such an alteration of MrpC binding affinity to P_{mrpC} could provide an explanation why ambruticin VS-3 leads to an inhibition of development.”

(l. 209-213) makes statements that are cryptic, unsupported, and not informative, so this should be deleted.

ANSWER of the authors: This was deleted.

(l. 244-246) is unclear. The data in Fig. S6 show that AskC is involved in starvation-induced developmental gene expression in the presence of ambruticin. Why to the authors say “but only AskAB is involved in development”? What does “additional entry points into the GRN that might be mirrored” mean? Presumably, the GRN for chemically-induced sporulation is different from that for starvation-induced multicellular development. Related to this, what does “AskAB are essential for development whereas AskABC is essential for sporulation what is highlighted by different colors.” mean in the Fig. 5b legend (l. 701)?

ANSWER of the authors: Due to the additional data now included in the manuscript, all these points have been deleted or specified.

(l. 260-262) and the model in Fig. 5c ignore the point mentioned above that ambruticin binds weakly to AskC in vitro. Are the authors proposing that ambruticin binds with higher affinity to AskABC? If so, this should be tested.

ANSWER of the authors: See answer to comment above.

Should “addressed” be “affected” on l. 300?

ANSWER of the authors: Yes, it should be “affected”. Changed accordingly.

Table S2 – reference 3 is missing and subsequent references are misnumbered.

ANSWER of the authors: Missing reference was added, misnumberig was changed.

Something is missing between “perfusing” and “data” on l. 413.

ANSWER of the authors: Has been added, thank you.

Reviewer #2 (Remarks to the Author):

The manuscript by Marcos-Torres et al describes how one myxobacteria species inhibits another's ability to undergo fruiting body development by a diffusible factor. They identify the causative agent, a secondary metabolite called ambruticin, which was previously shown to have antifungal activity. The potency of ambruticin is striking, 4 nM, suggesting this activity is physiologically relevant. Interestingly they also show that ambruticin induces sporulation by the classic glycerol induction pathway during non-starvation conditions. They then discovered three histidine kinases as playing a role in the ambruticin/glycerol pathways and that a key developmental regulator, MrpC, is regulated by ambruticin. Overall this study reports interesting findings and much of the experimental work is carefully done and impressive. Their findings also add depth and complexities to central aspects of the 'gene regulatory network' (GRN) controlling development in *Myxococcus xanthus*. However, there are missing pieces and how their findings fit into a cogent model still needs to be explored and resolved. Additionally, the text needs to be polished; there are numerous grammatical errors and the logic flow at times is difficult to follow. Some examples are given below.

General points:

Does ambruticin block development of other myxobacteria isolates/species? If so, this would strengthen the authors' argument that this activity is indeed physiologically/ecologically relevant.

ANSWER of the authors: This has not yet been excessively studied. Certainly, a knowledge about the spectrum of targets of ambruticin within other soil-living myxobacteria would be favorable and co-cultivations of such kind are ongoing. However, under laboratory conditions, the ability of myxobacteria to form fruiting bodies is frequently lost and therefore from time to time the "negative controls" are lost. A prominent case is *Chondromyces crocatus* Cmc5. Also in case of *M. xanthus*, it took a while to achieve reproducible FB formation. A first insight into a spectrum of target myxobacteria could be obtained when searching for myxobacteria encoding all kinases AskA, AskB and AskC. Amongst all published myxobacterial genomes, homologues of AskA, AskB and AskC could be found almost exclusively in the suborder of Cystobacterineae. At this, the kinase genes seemed to be highly conserved (query coverage above 99 % with a pairwise identity of 63.3 to 100 %). We think that we now have enough data to state that this is indeed physiologically/ ecologically relevant.

However, this question is similar to a comment of reviewer #1 and we want to comment on this as well: The extracellular concentration of ambruticin during a 6 days incubation period of *S. cellulosum* has been determined and is now included in the text/figures. The concentration is at least 80-fold higher than the 4nM used in the assays (see now in Fig.1). Up to date, nothing is known how this substance is released by the producer or how it is subsequently taken up by the target cell (*M. xanthus*). This is subject of future work and will be addressed in additional studies. In both cases, it is likely that this transport is not achieved by diffusion. All observations

account for ambruticin to be the causative agent: Only ambruticin-producers cause this phenotype in *M. xanthus* (see now also photographs of coinoculations of *M. xanthus* with non-producing myxobacterial strains in Fig. 1). The extracellular concentration of ambruticin exceeds the concentration of tested pure compound, and the observations made by addition of pure ambruticin equal the observations made during coinoculations with the ambruticin producing *S. cellulosum*.

To aid the reader, the chemical structure of ambruticin should be provided, preferably in Fig. 1.

ANSWER of the authors: The 2D structure has been added to Fig 1.

Table 1 and elsewhere, how is ambVS3 (ambruticin VS3) different than ambruticin and why are the results for MrpC binding different?

ANSWER of the authors: In all experiments, ambruticin VS-3 was used. This is the main compound produced by *S. cellulosum* and belongs to a number of derivatives of the ambruticin family of compounds. In the whole text “ambruticin” has been replaced by “ambruticin VS-3”. The binding for MrpC: Binding of MrpC to ambruticin VS-3 has been tested by SPR and MST. However, no clear binding could be observed (see manuscript). Therefore “n” for no binding in Table 1. In EMSA we could observe a significantly altered binding of MrpC to DNA comprising PmrpC upon addition of ambruticin VS-3 (Fig. 4a). The lowest concentration of ambruticin VS-3 triggering this effect in EMSA was listed in table 1 as an estimated minimum value of a KD [μ M] for an interaction of MrpC-PmrpC-ambruticin VS-3. Obviously, ambruticin VS-3 needs the presence of both, MrpC and the DNA comprising PmrpC to bind effectively and thereby increasing the affinity of MrpC to PmrpC. These findings have been more intensely described.

Lines 98-113 and Fig. S1 discusses/shows the domain architecture of various HK with HAMP domains, but the text lacks a clear explanation of what HAMP domains do and how they might function in these particular proteins. From the literature, HAMP motifs are typically found immediately downstream of transmembrane helices and serve as communication modules between the input and output domains of HK. This is not discussed or whether these particular HK have TM helices. Additionally, it is very unusual for proteins to have many (e.g. 16) HAMP motifs, which should also be discussed.

ANSWER of the authors: A respective explanation and information of their known functions has been added to the text. In addition, it has been clarified now that group III HHK share two extraordinary features: A lack of transmembrane helices and an unusual N-terminal region comprising a large number of HAMP domains (poly-HAMP). See also respective references that were added to the manuscript.

In Fig. S2, explain why some of the titration curves go down while others go up.

ANSWER of the authors: MST makes use of the thermophoretic properties of one fluorescently labeled protein to decipher whether an interaction with an additional protein occurs. An interaction leads to altered fluorescence signal intensity due to a change in conformation what is

measured. This can either lead to a decreasing or increasing fluorescence signal depending on the unique properties of each labeled protein.

Fig. 4a. Legend states all bands were identified by MALD-Tof, therefore please state the identity of the other protein band running at 55 kDa.

ANSWER of the authors: These bands correspond to AhpC (MXAN_1564), a peroxiredoxin. This information is now included in the figure legend (Fig 4).

Fig. 4a,b. The impact of Amb addition on MrpC is rather potent and intriguing. What happens if glycerol is instead added to cells in these experiments?

ANSWER of the authors: Indeed one could add glycerol to growing *M. xanthus* and compare proteins which are able to bind to PmrpC with those binding in presence of ambruticin VS-3 (Fig. 4a). This would be helpful when further investigating possible differences of both chemical inducers. However, as we focused on ambruticin VS-3, we think it is reasonable at this time not to add any further experiments addressing this as this might go beyond the scope of the manuscript. In future work to decipher the differences in the action of both chemical inducers, this should be included. However, adding the required high concentrations of glycerol will most likely change growth in general and thus cause difficulties in comparing data. Adding glycerol at respective concentrations to an EMSA (regarding Fig. 4b) will not work, as these high amounts of glycerol will lead to a strong retardation of all protein/DNA filled in the pockets and migration will strongly be reduced.

Fig. 4c. The finding that AskB binds specifically to the mrpC promoter is interesting. By extension from the results shown in

Fig. 4ab, what happens to the AskB DNA binding activity when ambruticin is added to cells prior to extract preparation? Also, when looking at the Fig. 1S cartoon structure of AskB it is not clear where a DNA binding domain could be and, moreover, as discussed, this is a unique finding that a HK binds DNA. It would therefore be informative to do structure-function studies to identify the DNA binding domain within AskB. Finally, the competitor DNA (Pmxan_0552) was used at equal amounts; what happens when competitor DNA is used in large excess, e.g. 100-fold?

ANSWER of the authors: The interaction of AskB with DNA comprising promoter PmrpC has been performed using heterologously expressed and purified AskB. So no cell extracts were involved in this EMSA (Fig. 4c). Proteinaceous extracts were only involved in a pull-down experiment in order to identify DNA-binding proteins binding to PmrpC in presence and absence of ambruticin (Fig 4a for MrpC).

Regarding Fig.1 S (domain organization of Ask kinases): The cartoon has been modified and now gives a comprehensive overview of predicted domains in these proteins. In the extended N-terminal parts of both proteins, AskA and AskB, numerous HAMP domains can be predicted using SMART (Simple Modular Architecture Research Tool), EMBL Heidelberg. Nevertheless, additional domains can be found in these N-termini but are overlapping with the predicted

HAMP domains. In both, AskA and AskB, numerous putative partial HTH_6_RpiR domains (helix-turn-helix) as well as partial putative SpoIIID-like domains comprising HTH_DEOR_1 domains could be found. These domains occur in a paired manner. Such domains cannot be found in AskC or Nik1. We think that one or more of these domains (HTH_6 and SpoIIID or both) mediate the DNA binding ability of AskB. This is now also mentioned in the text. We could now prove binding of AskA to PmrpC in the additional pull-down experiments (now included in the manuscript). As AskA also possesses such predicted domains in its N-terminus, they could be a good explanation for the observed binding to DNA. The altered number and organization of such domains in AskA might be a first hint to specificity or affinity. This could help to explain why we do not see AskA binding to DNA in EMSA but in pull-down experiments. Nevertheless, to us it would make sense if AskA would indeed be able to bind to DNA as phenotypes and promoter activity of PmrpC in Δ askA and Δ askB resemble the wild type. This is not true for the double mutant Δ askAB. One of the two has to be functional to enable development. We will address the question of DNA binding of the kinases in a follow up study as this represents another finding with implications on other fields of research. Regarding the competitor DNA Pmxan_0552 that was used in an equal amount compared to PmrpC-DNA: We decided to perform an additional EMSA using AskB, both differently labeled DNA fragments, and to add a 300-fold excess of non-labeled PmrpC-DNA. We clearly show that the addition of excess non-labeled DNA abolishes the shift in the EMSA experiment.

Line 193: Since the three kinases (AskABC) bind to MrpC and FruA, please discuss what is known about these latter proteins being phosphorylated.

ANSWER of the authors: This has been added and discussed now.

Since ambruticin is an antifungal agent the question arises whether it has antibacterial activity on *M. xanthus*. In this regard, does ambruticin block the growth of an askC mutant (which is refractive to sporulation induction)?

ANSWER of the authors: Like mentioned in the manuscript, the mutant Δ askC is insensitive towards ambruticin VS-3. This means ambruticin VS-3 is not affecting growth either.

Prior work by O'Connor & Zusman showed that induction of the glycerol spore pathway involves chromosomal induction of beta-lactamase in response to agents that damage the peptidoglycan, as well as, occurring naturally during fruiting body development. The developmental transition from vegetative cells to spores obviously involves major morphological changes and re-organization of the peptidoglycan, and agents that damage the peptidoglycan apparently induce sporulation independently of starvation. This work is relevant and accordingly should be discussed.

ANSWER of the authors: This has now discussed respectively.

Some of the experiments and interpretations need better explanations. For example, the cartoons in Fig. S5 seem to indicate protein fusions were created where the REC or REC3 domains were placed up or downstream of the p18/25 domains. If this is correct it should be described in the

legend and/or Result section and why it was done.

ANSWER of the authors: All interpretations in the text have been revised where necessary according to this reviewer's suggestion. The mentioned cartoons in Fig. S5 have been clarified in both, the legend and the Method section.

Minor points:

The last sentence of the abstract is confusing and needs to be re-worded.

ANSWER of the authors: This has been deleted.

Line 40: "AEC-signals" is a confusing abbreviation because it involves three signals, but reads as one. Change to "A, E and C-signals" or in a similar way to make the distinction.

ANSWER of the authors: The wording has been changed.

Line 42: change "influencing" to "controlling"

ANSWER of the authors: The wording has been changed.

Line 55: change "effective" to "potent"

ANSWER of the authors: The wording has been changed.

Line 56: add commas before and after "AskABC"

ANSWER of the authors: The wording has been changed.

Line 586: after "48" add "h"

ANSWER of the authors: Hours has been added.

The second paragraph, lines 114-130, is terse and difficult to follow. More rationale and data interpretation is needed, e.g. provide a preface to AskA and AskB binding to each other.

ANSWER of the authors: The paragraph was adjusted due to the characterization of additional mutants $\Delta askAC$, $\Delta askBC$ and $\Delta askABC$. Consequently, the section has been rearranged and both, rationale and data interpretation was added.

Line 133, after "glycerol" add "in rich medium" or "under starvation conditions" or re-write sentence, so the results can be clearly understood.

ANSWER of the authors: The sentence has been modified.

Line 175: insert "reporter" after "all" to qualify statement.

ANSWER of the authors: “reporter” has been added.

Line 186: The statement “is the main causal effect” is too strong as other cellular factors could also play a role. Change to something like “could provide an explanation for why ambruticin...”

ANSWER of the authors: The statement has been modified to: “Such an alteration of MrpC binding affinity to PmrpC could provide an explanation for why ambruticin VS-3 leads to an inhibition of development.”

Fig. S4 legend, line 113-114, please clarify sentence. What does “...any other protein in the absence of DNA...” mean?

ANSWER of the authors: The sentence has been reworded. No DNA bearing the promoter PmrpC was added to the buffers during the SPR / MST measurements. This is important as we observed a significantly altered binding of MrpC to its promoter in EMSA upon addition of ambruticin VS-3 at very low concentration. In SPR or MST we could not see binding of ambruticin to MrpC indicating the importance of the presence of DNA. The question has accordingly been addressed in the text.

Line 207-8: change “...we performed initial additional EMSA assays, also including the kinases” to “we performed preliminary EMSA assays that also included the kinases.” The next three sentences should be deleted or re-worded because they are vague/not informative.

ANSWER of the authors: The paragraph has been re-worded.

REVIEWERS' COMMENTS

Reviewer #1 (Remarks to the Author):

As noted by both previous reviewers, the authors have some very interesting results. Importantly, they have done a much better job of articulating the potential biological significance of the interspecies interaction they have discovered. The authors have also performed some of the experiments requested by the reviewers, and this has made the work more complete. In particular, the addition of ambruticin non-producers to Fig. 1a strengthens the evidence that ambruticin causes the observed effect. The new Fig. 1b documenting considerable production of ambruticin under unspecified conditions is not very helpful as presented (see comment below). The authors completed the genetic analysis by constructing and characterizing double and triple mutants, making a nice addition, although the results need to be presented a bit more thoroughly. The authors also added DNA-protein and protein-protein pull-down experiments to Fig. 4 (panels d and e, respectively). Fig. 4d is worthwhile since it shows that AskA, as well as AskB, binds to the *mrpC* promoter region. It also shows that AskC comes along with AskA and/or AskB, and that neither MrpC nor ambruticin changes these interactions. Fig. 4e is not as worthwhile and should be moved to supplemental. It confirms pairwise interactions observed by other methods. Importantly, it does not confirm a three-member AskABC complex as the authors claim (see comment below). Below and in a marked manuscript (mm), I strongly recommend numerous revisions to improve the writing of the manuscript.

I. 17 – “whether this is a natural process remains elusive” is puzzling without more introduction. See marked manuscript (mm) for suggested wording. The mm also suggests other wording and grammatical changes in the Abstract (and elsewhere), some of which are explained below.

I. 27 – the pull-down experiments do not show that the three kinases interact simultaneously. The data in Fig. 4e do not rule out separate interactions of AskA and AskB with AskC, because AskC is clearly in excess of AskA and full-length AskB (an abundant AskB fragment that binds AskC is also present, but this only confounds the interpretation). In this case, gel filtration (after release of AskC from the beads) would be required to show the existence of a three-membered complex in vitro.

I. 31 – while the effects of ambruticin and glycerol are similar, they are not equal in all respects.

I. 63 – “directly” overstates the evidence, which is based on effects of mutations on reporter activity in vivo. The effects could be indirect in vivo, even though the authors show kinase binding to the *mrpC* promoter region in vitro. They would need to map the kinase binding sites, make mutations that interfere with binding, and observe effects on reporter activity, to convince this reviewer the effects are direct.

Fig. 1b is potentially an important addition to the manuscript, but it needs to be explained much more carefully. Presumably, So ce 10 was grown in liquid. What was the culture density at the times the ambruticin concentration was measured? How was it measured? Is the concentration high enough to explain the inhibition of *M. xanthus* development on starvation agar plates (Fig. 1a), assuming ambruticin diffuses freely in the agar? Most importantly, what happens to ambruticin production when So ce 10 is starved? The answers to these questions impact whether one believes “Strain So ce10 produces sufficient amounts of ambruticin VS-3. The production titer of ambruticin VS-3 significantly exceeds the minimal concentration needed for inhibition of FB formation when using purified ambruticin VS-3 (see above).” on I. 108-111.

I. 87 and elsewhere – too many significant figures in many cases. For example, on I. 87, 0.35 should be 0.4 since the standard deviation indicates confidence only to 0.1, not 0.01, and likewise 4.37 should be 4 and 0.62 should be 1 since the standard deviation indicates confidence only to 1, not 0.1.

I. 157 – “to some extent” should be “efficiently” based on the data in Fig. 2c.

I. 163 – “equals” was very confusing. See mm. Note that the askAB double mutant with no additives is very different from DK1622 with glycerol or ambruticin in terms of germination efficiency of the pseudospores (Fig. 2c).

I. 186 – were domains of AskB and AskC tested for interaction using bacterial two-hybrid analysis? Why or why not? The sentence that begins “To elucidate...” suggests that this interaction would be tested in vivo.

Table 1 – the standard deviation of each measurement should be used to determine the number of significant figures to report in the mean and standard deviation (see comment on I. 87 above). For example, the mean KD for the AskC-ambVS3 interaction should 300 and the standard deviation should be 200, both in Table 1 and on I. 193 (where the numbers don’t match Table 1).

Fig. 2c – the result for the askAC mutant in the presence of glycerol and ambruticin is opposite the expectation from Fig. 2a and 2b, and the text on I. 205, which indicate “this strain is insensitive towards ambruticin VS-3 but not glycerol”

I. 216 – the authors should add a sentence after “under participation of AskC”. I suggest – “However, it is worth noting that in the absence of AskC and AskB, AskA is sufficient for a partial response to ambruticin and glycerol, and in the absence of AskC and AskA, AskB is sufficient for a partial response to glycerol, although in both cases the pseudospores formed by the double mutants germinate more efficiently than those formed by DK1622.”

I. 277 – why did the authors focus on ambruticin and not glycerol? A short explanatory sentence after the first sentence of the paragraph would be helpful.

Fig. 4 – the legend is excessive. There are too many experimental details and too much presentation of results. See mm for suggestions.

Fig. 4c – does AskB shift the mrpC promoter region DNA to the well of the gel? If so, the authors should mention this and suggest that a large complex forms.

I. 360 – “the binding of MrpC and the kinases seemed not to be mutually exclusive.” I agree, because the proteins are in vast excess over the DNA. To show this rigorously though, the AskA, AskB, and MrpC binding constants would need to be determined, so “seemed” is appropriate.

I. 454 – do the authors mean PmrpC-DNA? All three Ask kinases interact with MrpC.

I. 861 – specify the duration of chemical induction. 6 h?

Fig. 4d, lane 7 – should “+” be “-” for MrpC?

Fig. 5 – the legend is excessive. Some parts are redundant with the text. See mm for suggestions.

Fig. S1 legend I. 56 – the consensus sequence of HTH_DeoR_1 and HTH_6_RpiR should be shown in panel b. The 11 sequences are quite different from those shown in red, so the stated matches to consensus sequences cannot be verified from the figure.

Fig. S2, S3, and S4 – same comment about significant figures as for I. 87 and Table 1 above.

The authors state in their response to Reviewer #1 comment 3 that “The mrpC mutant has already been tested and did not sporulate, hence MrpC is required for ambruticin-induced sporulation (data

not shown in the manuscript but now mentioned).”, but I could not find this in the manuscript.

The authors did not perform the ChIP experiment requested by Reviewer #1 comment 6. I agree this is not a trivial request. The authors would need antibodies to AskB or an epitope-tagged version of AskB (ideally AskA as well based on the new results in Fig. 4d). Given that both AskB and AskA interact with MrpC, and MrpC interacts with the mrpC promoter region, the experiment would need to be done with an mrpC mutant in order to demonstrate DNA binding by the kinases *in vivo* (as opposed to cross-linking of kinase to MrpC bound to DNA).

Reviewer #2 (Remarks to the Author):

The authors have done a commendable job responding to reviewer concerns. Their work has uncovered complex inter-species interactions at a molecular level and has shed important new light on details of the development program in the model organism *M. xanthus*. I support acceptance of this publication with a few minor edits noted below.

Line 72: delete “up”

Lines 79: change “is” to “are”

Line 86: delete “per se”

Line 95: after “known” insert “to occur in the presence...”

Line 108-111: These two sentences can be merged into one for improved clarity.

In Fig. S1 legend or main text the E-values or the like should be given for the HAMP, GAF, HHK, HTH_6, SpoIID and other domains found in Amb07, AskA and AskC.

Fig. 4d is not cited in the manuscript.

There is inconsistencies in the use of italics in gene designations. For example, line 426 uses italics, but line 427 does not (*delta askABC*).

Lines 698-699, 787. Correct full capitals in these references.

Response to the reviewers

Reviewer #1 (Remarks to the Author):

As noted by both previous reviewers, the authors have some very interesting results. Importantly, they have done a much better job of articulating the potential biological significance of the interspecies interaction they have discovered. The authors have also performed some of the experiments requested by the reviewers, and this has made the work more complete. In particular, the addition of ambruticin non-producers to Fig. 1a strengthens the evidence that ambruticin causes the observed effect. The new Fig. 1b documenting considerable production of ambruticin under unspecified conditions is not very helpful as presented (see comment below). The authors completed the genetic analysis by constructing and characterizing double and triple mutants, making a nice addition, although the results need to be presented a bit more thoroughly. The authors also added DNA-protein and protein-protein pull-down experiments to Fig. 4 (panels d and e, respectively). Fig. 4d is worthwhile

since it shows that AskA, as well as AskB, binds to the mrpC promoter region. It also shows that AskC comes along with AskA and/or AskB, and that neither MrpC nor ambruticin changes these interactions. Fig. 4e is not as worthwhile and should be moved to supplemental. It confirms pairwise interactions observed by other methods. Importantly, it does not confirm a three-member AskABC complex as the authors claim (see comment below). Below and in a marked manuscript (mm), I strongly recommend numerous revisions to improve the writing of the manuscript.

l. 17 – “whether this is a natural process remains elusive” is puzzling without more introduction. See marked manuscript (mm) for suggested wording. The mm also suggests other wording and grammatical changes in the Abstract (and elsewhere), some of which are explained below.

Answer of the authors: The wording has been changed accordingly.

l. 27 – the pull-down experiments do not show that the three kinases interact simultaneously. The data in Fig. 4e do not rule out separate interactions of AskA and AskB with AskC, because AskC is clearly in excess of AskA and full-length AskB (an abundant AskB fragment that binds AskC is also present, but this only confounds the interpretation). In this case, gel filtration (after release of AskC from the beads) would be required to show the existence of a three-membered complex in vitro.

Answer of the authors: This has been changed in the abstract and elsewhere in the text.

l. 31 – while the effects of ambruticin and glycerol are similar, they are not equal in all respects.

Answer of the authors: The wording has been changed accordingly.

l. 63 – “directly” overstates the evidence, which is based on effects of mutations on reporter activity in vivo. The effects could be indirect in vivo, even though the authors show kinase binding to the mrpC promoter region in vitro. They would need to map the kinase binding sites, make mutations that interfere with binding, and observe effects on reporter activity, to convince this reviewer the effects are direct.

Answer of the authors: “directly” has been deleted.

Fig. 1b is potentially an important addition to the manuscript, but it needs to be explained much more carefully. Presumably, So ce 10 was grown in liquid. What was the culture density at the times the ambruticin concentration was measured? How was it measured? Is the concentration high enough to explain the inhibition of *M. xanthus* development on starvation agar plates (Fig. 1a), assuming ambruticin diffuses freely in the agar? Most importantly, what happens to ambruticin production when So ce 10 is starved? The answers to these questions impact whether one believes “Strain So ce10 produces sufficient amounts of ambruticin VS-3. The production titer of ambruticin VS-3 significantly exceeds the minimal concentration needed for inhibition of FB formation when using purified ambruticin VS-3 (see above).” on l. 108-111.

Answer of the authors: First, we declare that we are absolutely convinced to provide sufficient data to clearly say that So ce 10 produces sufficient amounts of ambruticin VS-3. This is based on the following facts:

1st: Only strains producing ambruticin are able to cause the described effect. All other strains used do not even possess a respective biosynthetic gene cluster.

2nd: We did show that ambruticin VS-3 as a pure compound in very low concentration is able to cause the effect. This is true for both, in liquid culture (100nM) and growth on agar (4nM). Actually this finding alone is evidence enough that ambruticin is the causative agent! In agar, ambruticin VS-3 is readily able to diffuse (this was already indicated by the coincubation of So ce 10 and *M. xanthus* and was observed when using filter discs treated with pure ambruticin VS-3 on top of the agar instead of mixing ambruticin VS-3 into the agar (not shown in the manuscript)).

3rd: The inhibitory effect on FB formation of *M. xanthus* can also be achieved by other ambruticin derivatives produced by strain So ce10. Amongst these, ambruticin VS-1 and ambruticin F were also tested as pure compounds on agar plates (data not shown). Both derivatives were effective starting from 4nM, the lowest concentration tested.

4th: The concentration of ambruticin VS-3 was determined from agar plates (VY2-agar), the same type of agar used in the coincubation. Therefore, a defined amount of cells of So ce10 was spotted on top of the agar (same amount used like in the coincubations). At each time point, agar from 10 agar plates was harvested, lyophilized and extracted. The amount of ambruticin VS-3 was determined by HPLC-MS measurement. Peak areas corresponding to ambruticin VS-3 in the samples together with those obtained from a standard curve of pure ambruticin VS-3 were used to calculate the overall yield of ambruticin VS-3.

The procedure has been added to the method section.

l. 87 and elsewhere – too many significant figures in many cases. For example, on l. 87, 0.35 should be 0.4 since the standard deviation indicates confidence only to 0.1, not 0.01, and likewise 4.37 should be 4 and 0.62 should be 1 since the standard deviation indicates confidence only to 1, not 0.1.

Answer of the authors: The values have been adjusted to match to the number of significant figures.

l. 157 – “to some extent” should be “efficiently” based on the data in Fig. 2c.

Answer of the authors: This has been changed.

l. 163 – “equals” was very confusing. See mm. Note that the askAB double mutant with no additives is very different from DK1622 with glycerol or ambruticin in terms of germination efficiency of the pseudospores (Fig. 2c).

Answer of the authors: This has been changed to: “This result, together with the observed effect that the Δ askAB mutant was unable to form FB under starvation conditions in the absence of additives, similar to the wild type DK1622 in the presence of ambruticin VS-3 or glycerol (Figure 2a), led to the hypothesis...”

l. 186 – were domains of AskB and AskC tested for interaction using bacterial two-hybrid analysis? Why or why not? The sentence that begins “To elucidate...” suggests that this interaction would be tested in vivo.

Answer of the authors: Only the in vivo interactions of domains of AskA and AskC have been tested. This has been clarified.

Table 1 – the standard deviation of each measurement should be used to determine the number of significant figures to report in the mean and standard deviation (see comment on l. 87 above). For example, the mean KD for the AskC-ambVS3 interaction should 300 and the standard deviation should be 200, both in Table 1 and on l. 193 (where the numbers don’t match Table 1).

Answer of the authors: The standard deviation of each measurement has now been used to determine the number of significant figures.

Fig. 2c – the result for the askAC mutant in the presence of glycerol and ambruticin is opposite the expectation from Fig. 2a and 2b, and the text on l. 205, which indicate “this strain is insensitive towards ambruticin VS-3 but not glycerol”

Answer of the authors: The askAC mutant is able to form FB in presence of ambruticin (Fig 2a) and consequently myxospores are formed (Fig 2b). In presence of glycerol no FB are formed (Fig 2a) and consequently only pseudospores are formed (Fig 2b). There is the observation that the askAC mutant’s myxospores seem to germinate less efficiently than spores produced in absence of ambruticin (Fig 2c). We changed the wording of former line 205 into: “...with no additives. Regarding FB formation, strain Δ askAC was insensitive towards ambruticin VS-3 but not glycerol (Figure 2a, 2b). However, the myxospores produced in these FB in presence of ambruticin VS-3 seem to germinate less efficiently (Figure 2c).” (l. 201).

l. 216 – the authors should add a sentence after “under participation of AskC”. I suggest – “However, it is worth noting that in the absence of AskC and AskB, AskA is sufficient for a partial response to ambruticin and glycerol, and in the absence of AskC and AskA, AskB is sufficient for a partial response to glycerol, although in both cases the pseudospores formed by the double mutants germinate more efficiently than those formed by DK1622.”

Answer of the authors: The sentence has been added, it will for sure help the reader.

l. 277 – why did the authors focus on ambruticin and not glycerol? A short explanatory sentence after the first sentence of the paragraph would be helpful.

Answer of the authors: A short explanation has been added.

Fig. 4 – the legend is excessive. There are too many experimental details and too much presentation of results. See mm for suggestions.

Answer of the authors: The legend has been shortened.

Fig. 4c – does AskB shift the mrpC promoter region DNA to the well of the gel? If so, the authors should mention this and suggest that a large complex forms.

Answer of the authors: Yes, this is the case. The following has been added: “Obviously, a large complex was formed, as the P_{mrpC}-DNA was shifted to the well of the gel.”.

l. 360 – “the binding of MrpC and the kinases seemed not to be mutually exclusive.” I agree, because the proteins are in vast excess over the DNA. To show this rigorously though, the AskA, AskB, and MrpC binding constants would need to be determined, so “seemed” is appropriate.

Answer of the authors: The determination of these binding constants is part of future work.

l. 454 – do the authors mean PmrpC-DNA? All three Ask kinases interact with MrpC.

Answer of the authors: The passage has been changed to: “In each combination, the resulting complex is able to interact with MrpC and/or with PmrpC-DNA.”

l. 861 – specify the duration of chemical induction. 6 h?

Answer of the authors: yes, it has been added.

Fig. 4d, lane 7 – should “+” be “-” for MrpC?

Answer of the authors: Yes, changed.

Fig. 5 – the legend is excessive. Some parts are redundant with the text. See mm for suggestions.

Answer of the authors: The legend has been shortened.

Fig. S1 legend l. 56 – the consensus sequence of HTH_DeoR_1 and HTH_6_RpiR should be shown in panel b. The 11 sequences are quite different from those shown in red, so the stated matches to consensus sequences cannot be verified from the figure.

Answer of the authors: Only sequences of E.coli RpiR and B.subtilis DeoR/SpoIIID had been shown. The figure has been adjusted. Sequence logos depicting possible residues and their occurrences at each position for all three domains have been added. Respective residues in the 11 regions matching the sequence logo parameters have been highlighted.

Fig. S2, S3, and S4 – same comment about significant figures as for l. 87 and Table 1 above.

Answer of the authors: When necessary, the values have been adjusted respectively.

The authors state in their response to Reviewer #1 comment 3 that “The mrpC mutant has already been tested and did not sporulate, hence MrpC is required for ambruticin-induced sporulation (data not shown in the manuscript but now mentioned).”, but I could not find this in the manuscript.

Answer of the authors: See now line 272 ff.

The authors did not perform the ChIP experiment requested by Reviewer #1 comment 6. I agree this is not a trivial request. The authors would need antibodies to AskB or an epitope-tagged version of AskB (ideally AskA as well based on the new results in Fig. 4d). Given that both AskB and AskA interact with MrpC, and MrpC interacts with the mrpC promoter region, the experiment would need to be done with an mrpC mutant in order to demonstrate DNA binding by the kinases *in vivo* (as opposed to cross-linking of kinase to MrpC bound to DNA).

Answer of the authors: We already discussed this point, but thank you again for explaining the method. As already pointed out in the response to the reviewers comment in the first revision, we changed the wording respectively and always refer to *in vitro* data.

Reviewer #2 (Remarks to the Author):

The authors have done a commendable job responding to reviewer concerns. Their work has uncovered complex inter-species interactions at a molecular level and has shed important new light on details of the development program in the model organism *M. xanthus*. I support acceptance of this publication with a few minor edits noted below.

Line 72: delete “up”

Answer of the authors: This has been deleted.

Lines 79: change “is” to “are”

Answer of the authors: This has been changed.

Line 86: delete "per se"

Answer of the authors: This has been deleted.

Line 95: after "known" insert "to occur in the presence..."

Answer of the authors: This has been added.

Line 108-111: These two sentences can be merged into one for improved clarity.

Answer of the authors: The sentences have been modified also according to the suggestion of reviewer 1.

In Fig. S1 legend or main text the E-values or the like should be given for the HAMP, GAF, HHK, HTH_6, SpoIID and other domains found in Amb07, AskA and AskC.

Answer of the authors: Only sequences of E.coli RpiR and B.subtilis DeoR/SpoIID had been shown. The figure has been adjusted. Sequence logos depicting possible residues and their occurrences at each position for all three domains have been added. Respective residues in the 11 regions matching the sequence logo parameters have been highlighted.

Fig. 4d is not cited in the manuscript.

Answer of the authors: It is now cited in line 350.

There is inconsistencies in the use of italics in gene designations. For example, line 426 uses italics, but line 427 does not (delta askABC).

Answer of the authors: Line 426 mentions the genes themselves; in line 427, a strain designation is given.

Lines 698-699, 787. Correct full capitals in these references.

Answer of the authors: This has been corrected.